# Taxon-Specific Shifts in Bacterial and Archaeal Transcription of Dissolved Organic Matter Cycling Genes in a Stratified Fjord

Benjamin Pontiller,[a,b] Clara Pérez-Martínez,[a] Carina Bunse,[a,c,d] Christofer M. G. Osbeck,[a] José M. González,[e] Daniel Lundin,[a] Jarone Pinhassi[a]

[a]Centre for Ecology and Evolution in Microbial Model Systems—EEMiS, Linnaeus University, Kalmar, Sweden
[b]GEOMAR Helmholtz Centre for Ocean Research Kiel, Kiel, Germany
[c]Helmholtz-Institute for Functional Marine Biodiversity at the University of Oldenburg (HIFMB), Oldenburg, Germany
[d]Institute for the Chemistry and Biology of the Marine Environment, University of Oldenburg, Oldenburg, Germany
[e]Department of Microbiology, University of La Laguna, La Laguna, Tenerife, Spain

**ABSTRACT** A considerable fraction of organic matter derived from photosynthesis in the euphotic zone settles into the ocean's interior and, as it progresses, is degraded by diverse microbial consortia that utilize a suite of extracellular enzymes and membrane transporters. Still, the molecular details that regulate carbon cycling across depths remain little explored. As stratification in fjords has made them attractive models to explore patterns in biological oceanography, we here analyzed bacterial and archaeal transcription in samples from five depth layers in the Gullmar Fjord, Sweden. Transcriptional variation over depth correlated with gradients in chlorophyll *a* and nutrient concentrations. Differences in transcription between sampling dates (summer and early autumn) were strongly correlated with ammonium concentrations, which potentially was linked with a stronger influence of (micro-)zooplankton grazing in summer. Transcriptional investment in carbohydrate-active enzymes (CAZymes) decreased with depth and shifted toward peptidases, partly a result of elevated CAZyme transcription by *Flavobacteriales*, *Cellvibrionales*, and *Synechococcales* at 2 to 25 m and a dominance of peptidase transcription by *Alteromonadales* and *Rhodobacterales* from 50 m down. In particular, CAZymes for chitin, laminarin, and glycogen were important. High levels of transcription of ammonium transporter genes by *Thaumarchaeota* at depth (up to 18% of total transcription), along with the genes for ammonia oxidation and $CO_2$ fixation, indicated that chemolithoautotrophy contributed to the carbon flux in the fjord. The taxon-specific expression of functional genes for processing of the marine pool of dissolved organic matter and inorganic nutrients across depths emphasizes the importance of different microbial foraging mechanisms over spatiotemporal scales for shaping biogeochemical cycles.

**IMPORTANCE** It is generally recognized that stratification in the ocean strongly influences both the community composition and the distribution of ecological functions of microbial communities, which in turn are expected to shape the biogeochemical cycling of essential elements over depth. Here, we used metatranscriptomics analysis to infer molecular detail on the distribution of gene systems central to the utilization of organic matter in a stratified marine system. We thereby uncovered that pronounced shifts in the transcription of genes encoding CAZymes, peptidases, and membrane transporters occurred over depth among key prokaryotic orders. This implies that sequential utilization and transformation of organic matter through the water column is a key feature that ultimately influences the efficiency of the biological carbon pump.

**KEYWORDS** marine bacteria, metatranscriptomics, dissolved organic carbon, carbohydrate-active enzymes, peptidases, transporters, vertical depth gradients, stratification, *Nitrosopumilus*, fjord

Address correspondence to Jarone Pinhassi, jarone.pinhassi@lnu.se.

The authors declare no conflict of interest.

Major portions of the primary production in the photic zone—up to ~40% of the photosynthetically fixed carbon—is transported vertically in the form of particulate organic matter into the ocean's interior in a process referred to as the biological carbon pump (1). This sinking organic matter is degraded by heterotrophic bacteria via extracellular enzymes that remineralize large proportions to carbon dioxide (2). Sinking particles cross steep gradients in light, temperature, nutrients, and hydrostatic pressure (3). Commonly, density gradients limit mixing between water masses, which disrupts the connectivity of microbial communities and nutrient fluxes. Stratification thereby strongly influences both the microbial community composition and the ecological function of these communities (4). This is, for example, visible in the replacement of phototrophy genes dominating in surface waters by chemolithoautotrophy genes at depth (5–9). This suggests that a pronounced variability in the genetic repertoire of bacteria and archaea is involved in the processing and uptake of nutrients and organic matter throughout the water column in the open ocean.

In the ocean, the bulk, i.e., community level, extracellular enzymatic hydrolysis rates rapidly decrease from epipelagic to bathypelagic zones, whereas the per-cell rates increase, indicating an increased microbial reliance on high-molecular-weight dissolved organic matter (HMW-DOM) with depth (10, 11). Actually, the hydrolysis of HMW-DOM is considered the rate-limiting step in the marine carbon cycle (12), and bacteria secrete hydrolytic enzymes to utilize particulate organic matter and biopolymers (13), e.g., carbohydrate-active enzymes (CAZymes) and peptidases (PEPs), which cleave HMW-DOM into molecules smaller than ~600 Da that can be transported through the cell membrane (14). A few recent studies applied metagenomics to study the spatial and vertical distribution of CAZyme and PEP genes (15–18). For instance, analysis of 94 metagenome-assembled genomes (MAGs) from the Mediterranean Sea uncovered a pronounced depth-related taxonomic and functional specialization in degradation of polysaccharides dominated by *Bacteroidetes*, *Verrucomicrobia*, and *Cyanobacteria* (16). Zhao and colleagues (18) applied a multi-omics approach encompassing a broad spatial and vertical coverage, showing that both the abundance and diversity of dissolved enzymes excreted by particle-attached prokaryotes consistently increased from epi- to bathypelagic waters. Still, knowledge of the concrete expression of these enzymes by different taxa through the water column is limited.

Fjord ecosystems exhibit pronounced vertical gradients in physicochemical and biological conditions in qualitative aspects reflecting gradients observed in other stratified coastal and offshore marine waters. Due to the reduced scaling of conditions in space and time, fjords are at times referred to as model oceans (19–21). Kristineberg Marine Research Station, created in 1877 and located in the Gullmar Fjord, a sill fjord on the west coast of Sweden, is one of the oldest marine research stations in the world. The extensive knowledge of the physical, chemical, and biological oceanography in the fjord provides a solid background against which to determine aspects of the microbial oceanography (22). Therefore, to obtain novel mechanistic knowledge of the functional degradation of biopolymers with depth and the microbial taxa that produce the required enzymes, we applied environmental metatranscriptomics to investigate the expression of polymer-degrading enzyme systems (i.e., CAZymes and PEPs). As we aimed at studying the combined responses of both degradation and uptake of ecologically important DOM compounds or nutrients in this system, we included membrane transporters in the analysis. We hypothesized that potential divergence in CAZyme, PEP, and transporter expression would be associated with shifts in dominance of transcription levels among taxa across depths.

## RESULTS AND DISCUSSION

**Variability of biotic and abiotic parameters.** Seawater samples were collected in duplicate from five depth layers (2 to 5, 15 to 25, 50 to 55, 75, and 100 m) in July and September 2017 at station Alsbäck (bottom depth, ~120 m) in the Gullmar Fjord, Sweden (Fig. 1A and B). Analysis of conductivity-temperature-depth (CTD) profiles

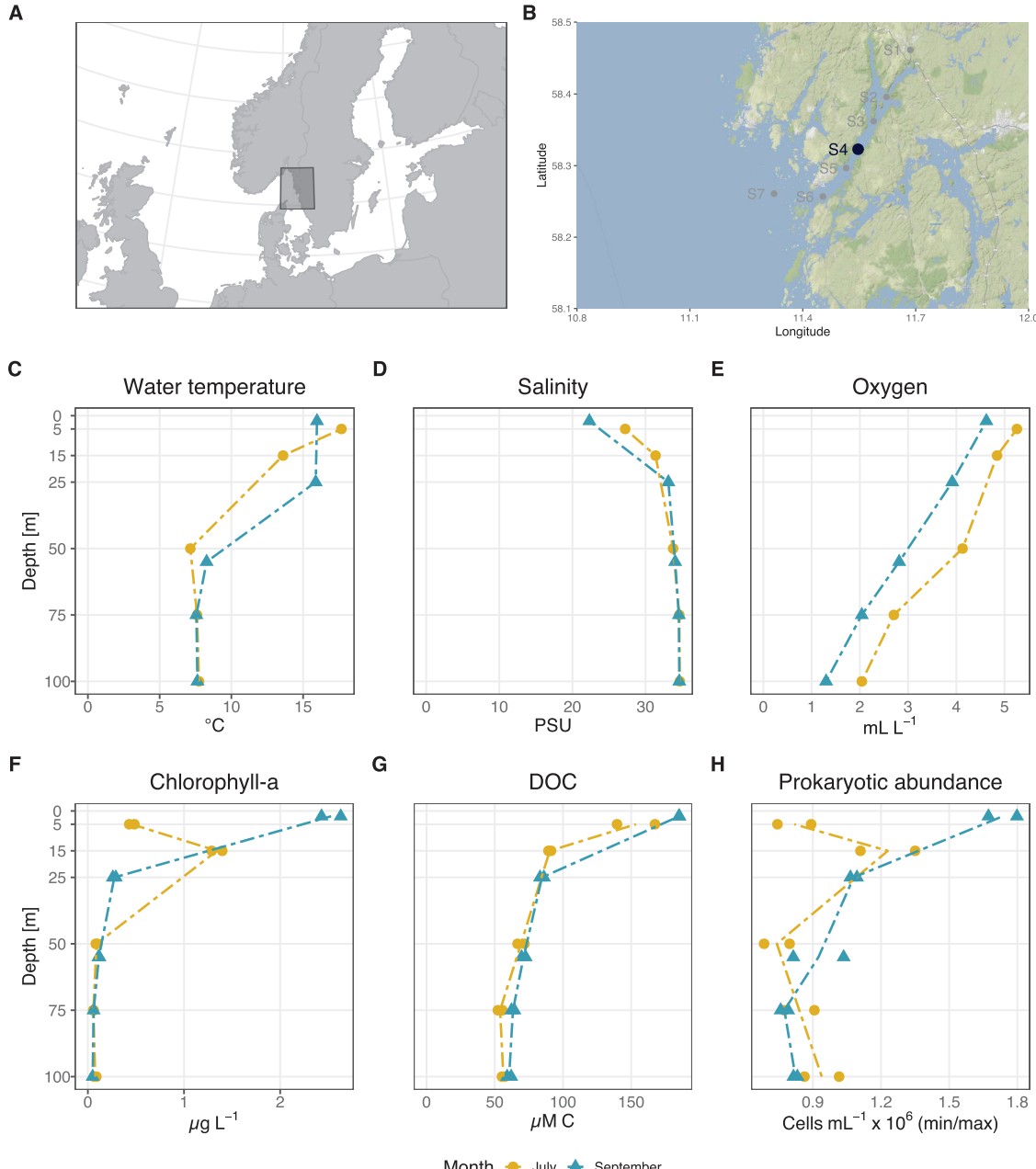

**FIG 1** Overview of study site and water column characteristics. (A and B) Locations of the Gullmar Fjord on the Swedish west coast (A) and the sampling site at station 4 (Alsbäck) with a bottom depth of 120 m (B). (C to H) Depth profiles of temperature (C), salinity (D), oxygen levels (E), chlorophyll *a* (B), dissolved organic carbon (DOC) (G), and prokaryotic abundance (H) in July and September.

showed a strong stratification of the water column in both July and September (Fig. 1C to E). The water temperature decreased rapidly from 13 to 17°C at the surface (2 to 25 m) to ∼8°C from 50 m and below. In September, temperatures were similar, at 16°C from 2 to 25 m and ∼8°C from 50 m (Fig. 1C). The surface salinity in July was 28 practical salinity units (PSU) and decreased to 22 PSU in September and at both times increased with depth to ∼36 PSU at 100 m (Fig. 1D), which corresponds to North Sea values (23). Dissolved oxygen levels steadily decreased with depth from oxic (∼4.5 to 5.2 ml liter$^{-1}$) at the surface to dysoxic (1 to 2 ml liter$^{-1}$) at the bottom (Fig. 1E). The vertical distribution of chlorophyll *a* (Chl *a*) concentrations differed substantially between months. In July, Chl *a* at the surface reached ∼0.5 $\mu$g liter$^{-1}$, increased to

$\sim$1.2 $\mu$g liter$^{-1}$ in the Chl $a$ maximum at 15 m, and decreased to $\sim$0.1 $\mu$g liter$^{-1}$ below 50 m depth (Fig. 1F). In September, concentrations were higher at the surface, reaching $\sim$2.8 $\mu$g liter$^{-1}$, but rapidly decreased to below $\sim$0.2 $\mu$g liter$^{-1}$ from 25 m downward (Fig. 1F). Interestingly, the variability between months seen for Chl $a$ was not reflected in the distribution of dissolved organic carbon (DOC) concentrations, which remained at maximum levels ($\sim$170 $\mu$M C) at the surface and decreased $\sim$3-fold to 100 m (Fig. 1G). Inorganic nutrients like $NO_3^+ + NO_2^+$ and $PO_4^{3-}$ increased from $<$1 $\mu$M to elevated levels at depth (up to 40 and 4.6 $\mu$M, respectively), although, for example, total nitrogen ($\sim$35 $\mu$M) remained relatively constant (see Fig. S1E in the supplemental material). The distribution of bacterial abundance resembled the vertical distribution of Chl $a$. Accordingly, bacterial abundance in July was generally low, at $\sim$0.9 $\times$ 10$^6$ cells ml$^{-1}$, except for a peak at $\sim$1.2 $\times$ 10$^6$ cells ml$^{-1}$ in the Chl $a$ maximum at 15 m depth. In September, bacterial abundance was highest in the surface, at $\sim$1.8 $\times$ 10$^6$ cells ml$^{-1}$, and steadily decreased with depth to $\sim$0.9 $\times$ 10$^6$ cells ml$^{-1}$ at 100 m (Fig. 1H).

The Chl $a$ concentrations measured here emphasize the mesotrophic nature of the Gullmar Fjord (22). The $\sim$2-fold difference between Chl $a$ concentrations in July and September was likely due to elevated grazing pressure in July (personal observation), in line with previous observations (22). In terms of physicochemical water column characteristics and nutrient dynamics during the time of sampling, the Gullmar Fjord compares to other fjord systems (19, 20). Still, we recognize that our current study is limited to a single location in the Gullmar Fjord, which, although sampled at two ecologically relevant time points, does not cover the full range of (a)biotic gradients along the extension of the fjord or how they change during a full year in this or other fjord systems.

**Overall patterns in prokaryotic community transcription by depth and month.** Bacterial and archaeal transcription changed with depth and between samplings (Fig. 2). A principal-component analysis (PCA) separated the widely spread surface samples (down to 25 m) from a tight cluster of deeper samples (50 to 100 m) (Fig. 2A). Detailed hierarchical cluster analysis showed four distinct clusters consisting of a July surface cluster (5 to 15 m), a September surface cluster (12 to 25 m depth), and a separate cluster consisting of a mix of deep-water samples (50 to 100 m depth) (Fig. S3A). Further, redundancy analysis (RDA) showed that the influence of sampling date (July or September) on bacterial and archaeal transcription was strongest in the surface layer and decreased with depth (Fig. 2A and B). These differences were mainly explained by $NH_4^+$ (13% of variation) and partly by Chl $a$ (9%), whereas depth variation was predominantly driven by DOC (12%) and $NO_3^- + NO_2^-$ (8%) (Fig. 2B; Fig. S2). This decrease in variability of transcription with depth was recently demonstrated for genes and taxa that showed diel oscillation in the surface but essentially diminished with depth through strong attenuation of light (24). Here, we show that this variability in transcription also applies to other gene systems, including genes coding for CAZymes, PEPs, and transporters (TPs).

*Alphaproteobacteria* and *Gammaproteobacteria* accounted for high portions (around 25 to 30%) of the prokaryotic community transcription in the surface waters and maintained elevated transcription ($\sim$11%) throughout the water column (Fig. 2C). *Cyanobacteria* (primarily *Synechococcus*) were most active in the upper water column (reaching $\sim$15% of community transcription). These patterns are largely in agreement with marine metagenomic and metatranscriptomic surveys from the open ocean, which typically report a dominance of *Picocyanobacteria* (*Prochlorococcus*), *Alphaproteobacteria* (primarily the SAR11 clade), and *Bacteroidetes* in the upper water column (6, 16, 25). The relatively high activity of *Verrucomicrobia* in July (reaching $\sim$45% of community transcription) is in line with a report from the Baltic Sea, where 16S rRNA gene analyses show that this taxon reaches higher relative abundances during summer coinciding with a dominance of *Cyanobacteria* (26). Metagenomic analysis of *Verrucomicrobia* shows that this taxon carries an extensive repertoire of CAZymes (e.g., alpha- and beta-galactosidases, xylanases, fucosidases, agarases, and endoglucanases), suggesting that they play a vital role in the

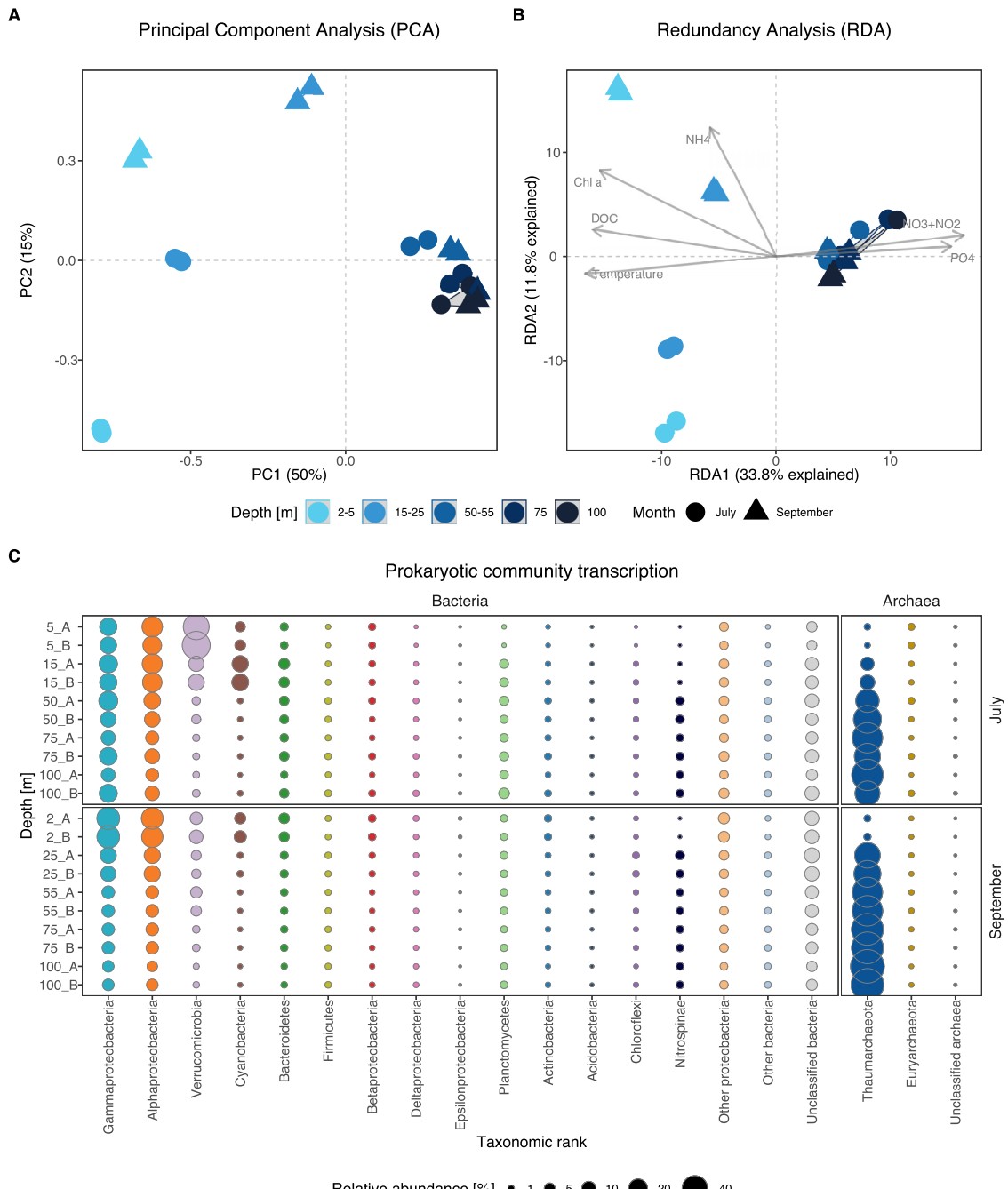

**FIG 2** Variability in prokaryotic community transcription across depth and between sampling dates. (A) Principal-component analysis (PCA) based on 82,842 quality filtered open reading frames (ORFs). (B) Redundancy analysis (RDA) constrained by environmental variables. (C) Taxonomic affiliation of the transcriptionally most abundant prokaryotic phyla. Note that *Proteobacteria* are shown at the class level.

degradation of phytoplankton-derived DOM and complex polysaccharides such as fucoidans (16, 27, 28).

Strikingly, *Thaumarchaeota* dominated transcription from 50 m downwards in July and from ~25 m in September, accounting for up to ~75% of total transcripts. Also, the functional contribution of *Nitrospinae* (~2.5%, <0.5% at 5 m) was noticeable in deeper water layers that were associated with lower oxygen concentrations (Fig. 2C and 1E). High proportions of *Thaumarchaeota* and *Nitrospinae* at depth have also been found in the northern Pacific and the Mediterranean Sea (6, 16, 25). *Thaumarchaeota*

are key players in the oceanic N cycle by oxidizing ammonia to nitrite (first step in nitrification) (7, 9) and generally increase in abundance with depth, accounting for up to 40% of total cells in the mesopelagic zone (29). *Nitrospinae*, in turn, are the most abundant and ubiquitously distributed nitrite-oxidizing bacteria (NOB) in the ocean (performing the second step in nitrification, oxidation of nitrite to nitrate) (8). In natural systems, *Thaumarchaeota* are typically ~10-fold more abundant than *Nitrospinae* (30). Thus, the transcriptional activity of *Thaumarchaeota* together with *Nitrospinae* below 50 m depth suggests an important contribution of these two groups to the N cycle in this stratified fjord and potentially to the C cycle, since both taxa are lithoautotrophs.

**Variation in bulk prokaryotic transcription of DOM transformation genes (CAZymes, PEPs, and TPs).** Averaged over the entire water column, *Bacteria* accounted for 97% of CAZyme (~0.3% of total transcripts) and ~90% of PEP (~3.4% of total transcripts) gene transcription. In contrast, *Archaea* devoted little transcriptional effort to CAZymes and PEPs but instead accounted for ~55% of TP transcription (~10.4% of total transcripts) (see Table S2 at https://doi.org/10.6084/m9.figshare.17029547). These patterns prompted us to determine the relationship between the three gene systems over depth (see Fig. S4 in the supplemental material). The prokaryotic CAZyme transcription relative to that of PEPs was up to 5- to 10-fold higher in the surface waters than at depths from 50 m and below; note in particular the high proportion of CAZymes at 15 m in July (Fig. S4A). In addition, we found that the proportion of CAZyme and PEP transcription relative to TPs decreased with depth (Fig. S4B and S4C), mainly because of the increasing dominance of *Archaea*. The higher proportion of CAZymes in the Chl *a* maximum layer agrees with enzymatic activity dynamics during phytoplankton blooms, particularly during bloom senescence, and in Chl *a* maximum layers in marine and limnic systems (11, 31). If the concentration of cleavage end products determines prokaryotic ectoenzyme activities, the relative increase in PEP transcription with depth suggests that prokaryotic communities increase their efforts to acquire organic N and P because exported DOM typically has high C-to-N ratios (32). This has been shown for cell-specific enzymes such as leucine aminopeptidases and alkaline phosphatases, which typically increase in activity with depth (10).

These pronounced patterns in depth distributions, in turn, inspired further analysis of the allocation of transcriptional efforts to CAZymes, PEPs, and TPs among the dominant prokaryotic orders (Fig. 3), in an attempt to mitigate the limitation of relative metatranscriptomic data (i.e., the abundance of gene transcripts calculated as percentage of the sequence library) to disentangle the extent or direction of change in gene transcription of complex natural communities with depth. This highlighted the disproportionate contribution of *Thaumarchaeota* in nutrient uptake over polymer degradation at depth. *Alphaproteobacteria* such as *Pelagibacterales* showed a relatively high and stable proportion of TP transcripts compared to degrading enzymes with depth and across sampling dates. A somewhat different trend was noticed for *Rhodobacterales*, which potentially engaged more in carbohydrate degradation in the upper water column but changed toward proteins with depth; this was more pronounced in September than in July (Fig. 3). Interestingly, a preference for carbohydrates in comparison to proteins was more accentuated in *Flavobacteriales* in the upper water column but shifted toward PEPs with depth, whereas *Cellvibrionales* invested consistently more in CAZymes than PEPs throughout the water column in July but showed a trend similar to that of *Flavobacteriales* in September, emphasizing their significant role in the turnover of carbohydrates in surface waters. *Alteromonadales*, in turn, contributed more to the expression of TPs than CAZymes and seemingly favored peptides over carbohydrates in subsurface layers (Fig. 3). *Synechococcales* showed a disproportionate investment in TPs compared to degrading enzymes from 2 to 15 m depth. However, PEPs and TPs increased over CAZymes from 50 m downwards with a disproportionate transcription of PEPs over TPs.

These results suggest that the prokaryotic community attempted to acquire N and P with depth, as seen by the disproportional investment in PEPs and TPs over CAZymes. However, the relative proportions of these gene systems varied substantially between orders throughout the water column and were surprisingly consistent

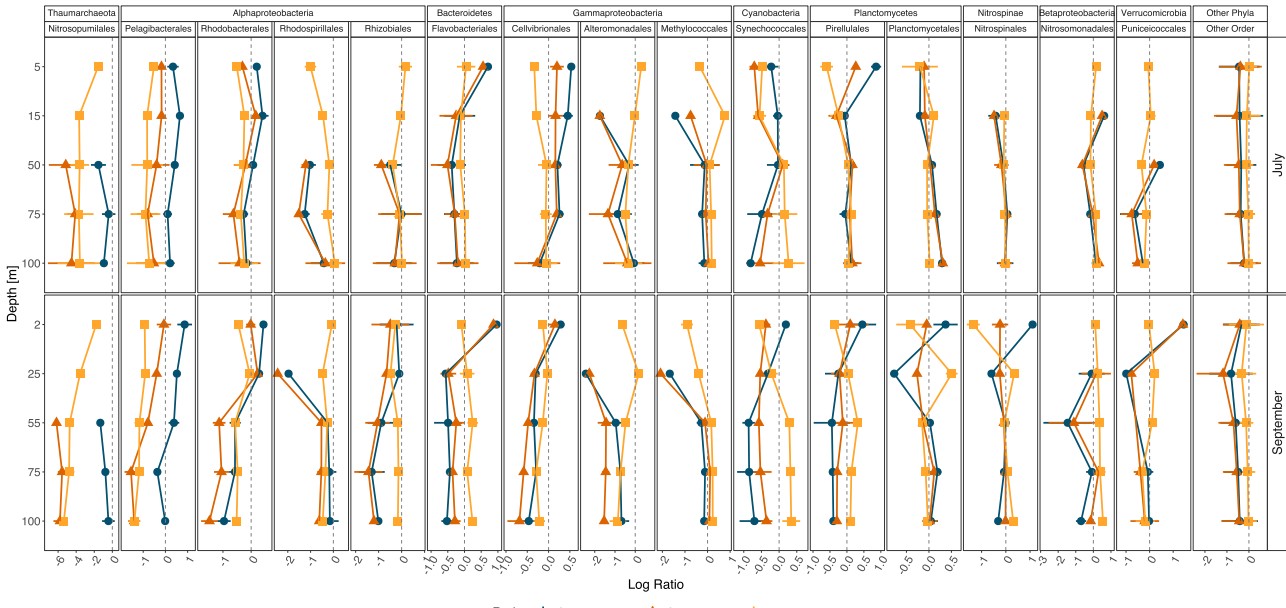

**FIG 3** DOM cycling gene transcription ratios of the most active prokaryotic orders across spatiotemporal gradients. Depicted are the mean log ratios ± SD ($n = 2$) of total CAZyme to PEP transcription within an order for a given depth and date (blue points). In addition, the mean log ratios ± SD ($n = 2$) of CAZyme to TP transcription (orange triangles) and PEPs to TPs (yellow squares) are shown for each order. For instance, *Alteromonadales* depicted a negative CAZyme/PEP log ratio, indicating that the relative proportion of PEP transcription was higher than the CAZyme transcription, in particular at 15 m depth. Moreover, negative CAZyme/TP log ratios suggested that the transcription of TPs was more important than that of CAZymes. The slightly decreasing trend of PEP/TP, from initially positive at 5 m depth to negative at 100 m depth, indicated that peptide degradation was more important in the upper water layer than uptake that increased with depth. Overall, *Alteromonadales* transcribed more PEPs and TPs than CAZymes. Note the different signs and values on the y and x axes.

between sampling dates for some orders (e.g., *Thaumarchaeota*, *Pelagibacterales*, and *Alteromonadales*). In addition, our findings show that a few bacterial groups, such as *Bacteroidetes*, *Cellvibrionales*, *Planctomycetes*, and *Verrucomicrobia*, showed relatively higher transcription of CAZymes than of transporters, particularly in the surface, suggesting that marine microbes regulate their transcriptional efforts in degrading enzymes and transporters in a depth-dependent manner. The variation in the ratios implies a crucial role of different bacterial and archaeal taxa in the processing of HMW-DOM, emphasizing their metabolic/transcriptional plasticity in transcribing CAZymes, PEPs, and TPs. Possible triggers of such variation could be changing elemental ratios of DOM and the desire to maintain a balanced stoichiometry. If confirmed in future analysis, this would have implications for the constraints of DOM cycling and microbial population dynamics.

**Divergence in CAZyme transcription.** The richness of expressed CAZymes differed significantly between different depth layers (analysis of variance [ANOVA]; $F_{4,15} = 14.87$, $P < 0.00004$) (Fig. 4A), with ~4-fold lower values in the surface (2 to 5 m depth) than in the deeper water layers (15 to 100 m depth) (Tukey; $P < 0.03$ to 0.0001). While both richness (Fig. 4A) and evenness (Fig. 4B) were fairly similar in July, the evenness in September at 25 m was lower than that in the deep.

The microbial community expression of CAZyme genes differed significantly with both depth (permutational multivariate ANOVA [PERMANOVA], $R^2 = 0.61$, adjusted $P$ value [$P_{adj}$] < 0.0003) and month (PERMANOVA, $R^2 = 0.1$, $P_{adj} < 0.001$) (Fig. 4C). Both PEP and CAZyme transcription in the surface layer (2 to 5 m) clustered with the samples from 15 to 25 m and distantly from the deeper samples (50 to 100 m) (Fig. S3B and C). Differences in clustering of samples between July and September were observed for the surface samples (2 to 5 and 15 to 25 m) but only minimally for the samples below 50 m depth (Fig. 4C; Fig. S3B and S3C).

The relative proportions of CAZyme classes were relatively stable throughout the water column (Fig. 4D), with the most abundant classes being glycoside hydrolases

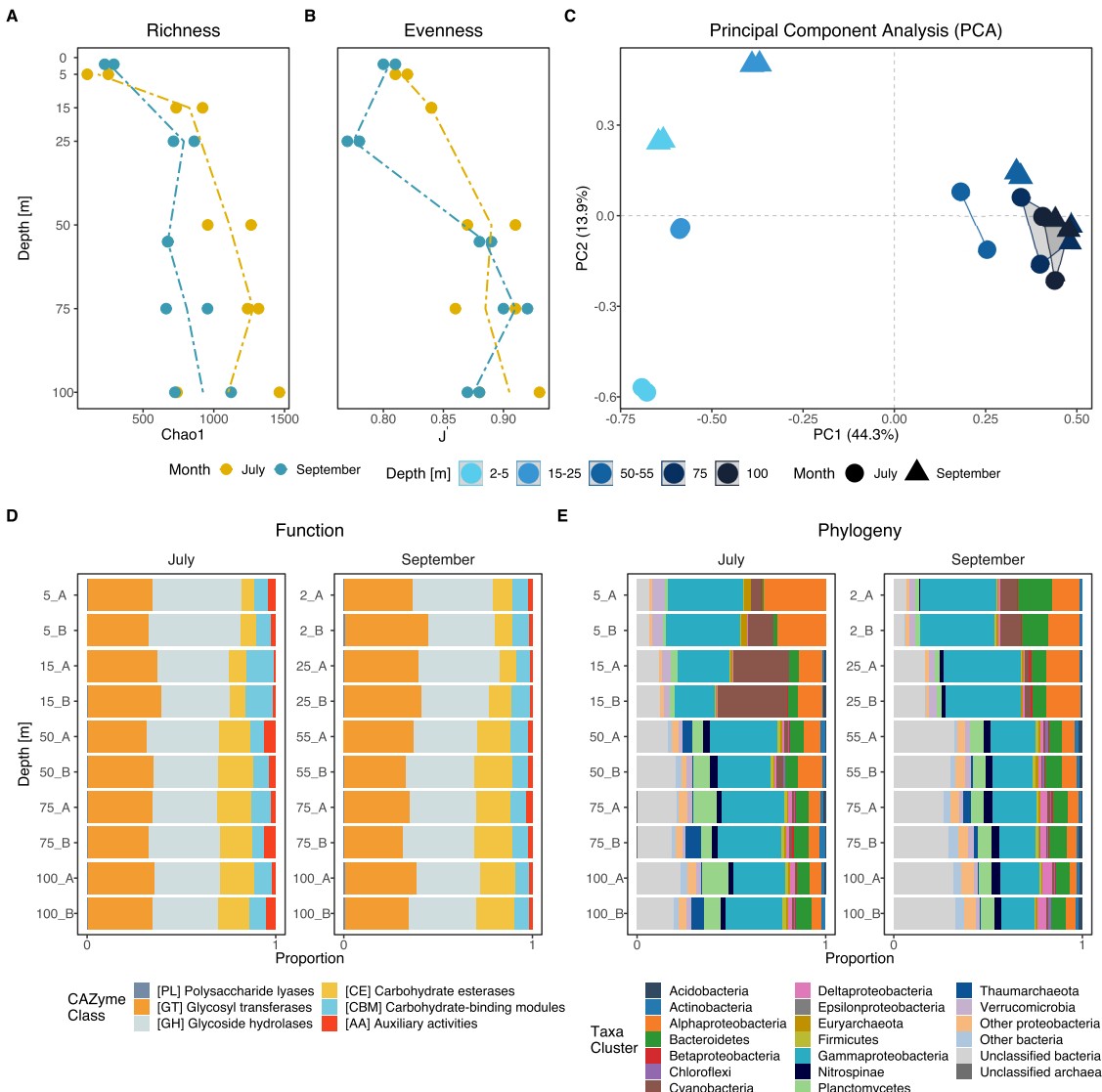

**FIG 4** Taxonomic and functional profiles of carbohydrate-active enzyme (CAZyme) transcripts. (A and B) The estimated richness (Chao1) (A) and evenness (J′) (B) in depth profiles of CAZymes at the ORF level. (C) Principal-component analysis (PCA) of 1,350 CAZyme-affiliated open reading frames (ORFs). (D and E) Functional transcripts at the CAZyme class level (D) and taxonomic affiliation of the most active phyla transcribing CAZymes (E). *Proteobacteria* are grouped at the class level.

(GHs; 37.9% ± 4.4%, *n* = 20) and glycosyltransferases (GTs; 35.6% ± 3.4%). This was surprising given that the taxonomic affiliation of these genes changed with depth (Fig. 4E). Nevertheless, Zhao et al. found a similar pattern among epi- to bathypelagic samples from the Pacific, Atlantic, and Southern Oceans (18). This suggests that the CAZyme class level is too coarse to identify functional changes across depths, which is in stark contrast to the distribution of CAZyme families (see below) (Fig. 4 and 5).

*Gammaproteobacteria* dominated CAZyme expression in both July and September, accounting for up to ~40% of CAZymes in the upper water layers and 10 to 30% from 50 m and below (Fig. 4E). *Alphaproteobacteria* were also abundant in transcription in the surface layers (up to ~30%), with expression levels down to 5% of CAZymes at depth. Interestingly, *Cyanobacteria* dominated in July in the subsurface Chl *a* maximum layer, where they accounted for ~34% of total CAZyme transcription. The significantly lower evenness observed in September (Fig. 4B) was likely the result of the *Gammaproteobacteria* being composed of a diverse set of taxa with a more uneven distribution of transcripts (Fig. 4B). Notably, from 50 m downward, *Planctomycetes*,

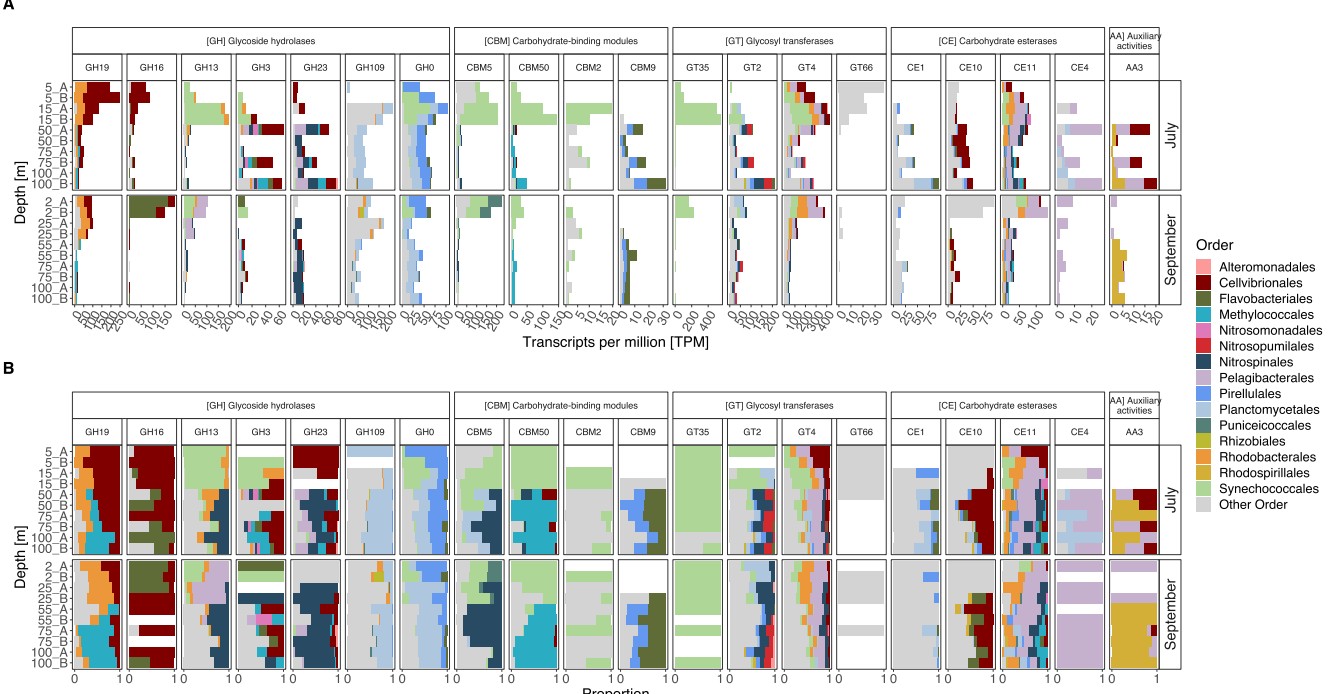

**FIG 5** Transcription of the 20 most abundant carbohydrate-active enzymes (CAZymes) grouped into CAZyme families and taxonomic orders. (A) Relative transcript abundance in transcripts per million (TPM); note different scales on *y* axes. (B) Proportions of CAZyme transcripts. Individual CAZyme families are further grouped into CAZyme classes (top facets).

*Nitrospinae*, and *Deltaproteobacteria* accounted for ~8%, 3.6%, and ~2.6% (both months) of total CAZyme transcription, respectively (Fig. 4E). *Verrucomicrobia* showed a relatively stable activity profile with depth, contributing 1 to 6% of CAZyme expression.

Already at the order level, there were pronounced differences in the transcription of the most abundant CAZyme families between months and depth (Fig. 5; Fig. S5). For instance, *Cellvibrionales* (e.g., *Halieaceae* family) and *Rhodobacterales* (e.g., *Rhodobacteraceae* family) dominated transcription of glycoside hydrolase family 19 (GH19) in the surface layer, whereas *Methylococcales* (e.g., *Methylococcaceae* family) dominated below 50 m depth (Fig. 5; Fig. S6). Actually, GH19 explained 18.5% of the variation in community transcription with depth and ~27% by month (see Table S3 at https://doi.org/10.6084/m9.figshare.17029559). GH19 consists primarily of chitinases that hydrolyze chitin, the primary component of cell walls in fungi and exoskeletons of crustaceans (33). Incidentally, we noticed a high abundance of copepods in microscopy samples from July that coincided with the low Chl *a*, in line with previous observations from this time of year (22). These findings suggest a crucial role of chitin and chitodextrins as resources for prokaryotic plankton. *Cellvibrionales* (e.g., *Cellvibrionaceae*, *Halieaceae*, and *Porticoccaceae* families) and *Flavobacteriales* (e.g., *Flavobacteriaceae* family) both transcribed GH16, with a larger proportion of *Flavobacteriales* in September (Fig. 5; Fig. S6). The GH16 family comprises laminarinases that allow bacteria to decompose the algal storage glucan laminarin (a beta-1,3-glucan) (31). *Cellvibrionales* and *Flavobacteriales* are typically abundant and active during phytoplankton blooms and likely fulfill different roles in the degradation of phytoplankton-derived organic matter (31). The CAZyme transcription patterns support pronounced intra- and interspecific niche partitioning in the turnover of organic matter over spatiotemporal scales.

*Synechococcales* (e.g., the genus *Synechococcus*) dominated transcription of GH13 at 15 m in July, with some contribution also from *Alphaproteobacteria* (e.g., *Rhodobacterales* and *Pelagibacteraceae* family) (Fig. 5; Fig. S6). GH13 contains, for example, alpha-amylases

that allow hydrolysis of the storage polymers glycogen and starch (34). In addition, *Synechococcales* showed a high transcription of glycosyltransferases (GT35 and GT4) and carbohydrate-binding modules (CBM50 and CBM5). The glycosyltransferase GT35 is associated with the synthesis of starch and glycogen (35), whereas GT4 is involved in the synthesis of cellular structures and energy storage (i.e., sucrose, mannose, and trehalose synthase) upon photosynthesis (15, 35). The carbohydrate-binding modules CBM50 and CBM5 are involved in chitin binding to facilitate the degradation of chitin or peptidoglycan (36). Interestingly, macroalgae and cyanobacteria express an extensive suite of CAZymes (e.g., cellulases, amylases, galactosidases), PEPs, and lipases (37). Given that *Cyanobacteria*, including *Synechococcales*, synthesize the polysaccharide glycogen as an internal carbon and energy storage compound (38), the relatively high expression of GH13 is likely associated with the utilization of internal glycogen sources rather than being a sign of extracellular degradation. Indeed, cyanobacterial genomes encode a relatively low number of genes for secretory enzymes (18). The strong transcriptional response in CAZyme transcription by *Synechococcales* (especially *Synechococcus*) highlights the importance of these genes in regulating glycogen metabolism across pronounced light and nutrient gradients by balancing organic carbon synthesis and degradation.

Two taxa stood out as having higher CAZyme transcription from 50 m and down: *Planctomycetales* and *Nitrospinales* (Fig. 4). *Planctomycetales* (e.g., *Planctomyces*, *Gimesia*, and *Rubinisphaera* genera) dominated transcription of $\alpha$-N-acetylgalactosaminidase (GH109) and glycoside hydrolases (G0, not yet classified in the CAZy database) (Fig. 5; Fig. S6). GH109 is involved in the degradation of bacterial cell walls (39) and has been shown to be abundant in metagenome-assembled genomes (MAGs) throughout the water column in the Mediterranean Sea (16) (Fig. 5). Since the cell walls of *Planctomycetales* lack the polymer peptidoglycan (40), the transcription of GH109 suggests a role in the degradation of cell walls of other bacteria. *Nitrospinales* increased transcription of GH23, CBM5, and GT2 with depth (Fig. 5). These CAZymes are involved in the degradation of peptidoglycan or chitin and the synthesis of chitin or cellulose (35, 39).

The transcriptional CAZyme responses by a diverse microbial community confirm the crucial role of the storage polysaccharide laminarin in fueling the heterotrophic carbon demand in this stratified fjord. Moreover, we noted strong transcriptional responses associated with the structural polysaccharides chitin and peptidoglycan. Remarkably, our results highlight a depth-layer-dependent functional partitioning in CAZyme transcription by well-known polymer degraders like *Cellvibrionales* and *Flavobacteriales*. Importantly, our analysis identified taxa like *Nitrospinales* (e.g., *Nitrospina* genus), *Nitrosomonadales* (e.g., *Nitrosopira* genus), and *Methylococcales* (e.g., *Methylococcaceae* family) that previously seem to have been overlooked in the context of polysaccharide degradation.

**Divergence in membrane transporter transcription.** The estimated richness of transcribed transporters increased with depth in July but was relatively constant in September (Fig. 6A). The evenness of transcribed transporters, however, was lower below 20 m depth, especially in September (Fig. 6B). Moreover, transcription of prokaryotic membrane transporter genes differed significantly with depth (PERMANOVA; $R^2 = 0.62$, $P_{adj} < 0.0003$) and sampling date (PERMANOVA; $R^2 = 0.08$, $P_{adj} < 0.002$) (Fig. 6C). As for overall transcription, cluster analysis of transporter transcription grouped samples into four distinct depth clusters (Fig. S3D). The largest variability in transcription was associated with the upper water layers, which clustered away from the samples from 50 m and below.

The most abundant transporter families were ammonium transporters (Amt; 47% $\pm$ 28%, number of samples [$n$] = 20), ATP-binding cassette transporters (ABC; 14% $\pm$ 6%, $n = 20$), and outer membrane receptors (OMR; 4% $\pm$ 3%, $n = 20$) (Fig. 6D). PCA of transporter transcription showed that these transporters contributed between 38% and 12% of variation in community gene expression with depth (PC1) and between 27% and 11% of variation by sampling date (PC2) (see Table S3 at https://doi .org/10.6084/m9.figshare.17029559). Generally, transporter transcription in the surface

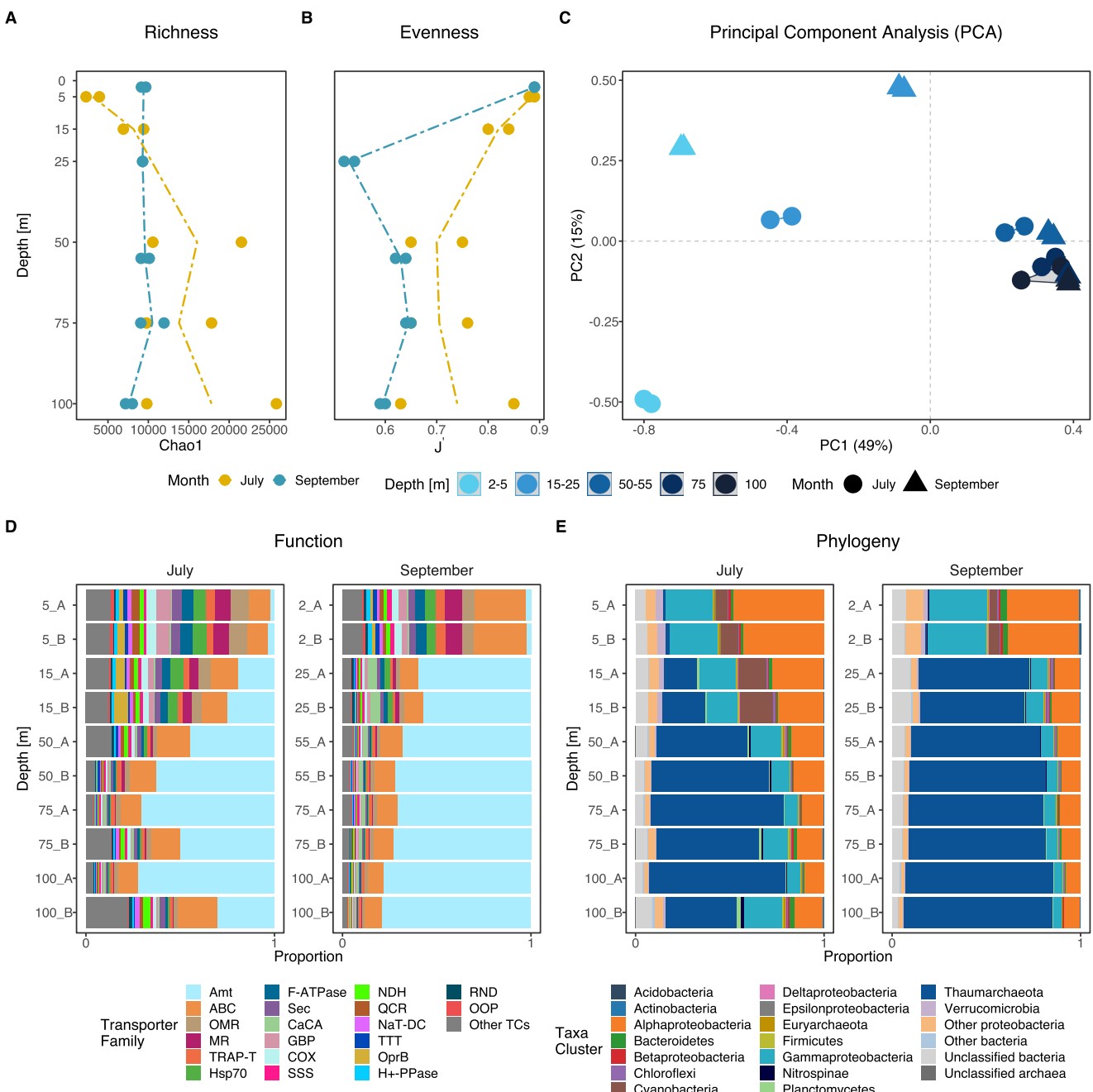

**FIG 6** Taxonomic and functional transcription of membrane transporters (TPs). (A and B) The estimated richness (Chao1) (A) and evenness (J′) (B) in depth profiles of TPs at the ORF level. (C) PCA of 25,395 TP affiliated open reading frames (ORFs). (D and E) Functional transcripts of the most abundant TP families (D), and taxonomic affiliation of the most active phyla transcribing TPs (E). Proteobacteria are shown at the class level. Abbreviations of transporter families: Amt, ammonium transporter channel; ABC, ATP-binding cassette; OMR, outer membrane receptor; MR, ion-translocating microbial rhodopsin; TRAP-T, tripartite ATP-independent periplasmic transporter; Hsp70, cation channel-forming heat shock protein 70; F-ATPase, $H^+$- or $Na^+$-translocating F-type, V-type, and A-type ATPase; Sec, general secretory pathway; CaCA, $Ca^{2+}$:cation antiporter; GBP, general bacterial porin; COX, $H^+$-translocating cytochrome oxidase; SSS, solute-sodium symporter; NDH, $H^+$- or $Na^+$-translocating NADH dehydrogenase; QCR, $H^+$-translocating quinol:cytochrome $c$ reductase; NaT-DC, $Na^+$-transporting carboxylic acid decarboxylase; TTT, tricarboxylate transporter; OprB, glucose-selective porin; $H^+$-PPase, $H^+$, $Na^+$-translocating pyrophosphatase; RND, resistance-nodulation-cell division; OOP, OmpA-OmpF porin.

layer was taxonomically similar to that of CAZymes. However, from 15 m depth and below, we noticed an exceptionally high contribution of *Thaumarchaeota* on both samplings, accounting for up to 79% of transporter transcription (Fig. 6E).

Our findings on membrane transporter transcription indicated that *Pelagibacterales* (e.g., "*Candidatus* genus Pelagibacter") and *Rhodobacterales* (e.g., *Rhodobacteraceae*

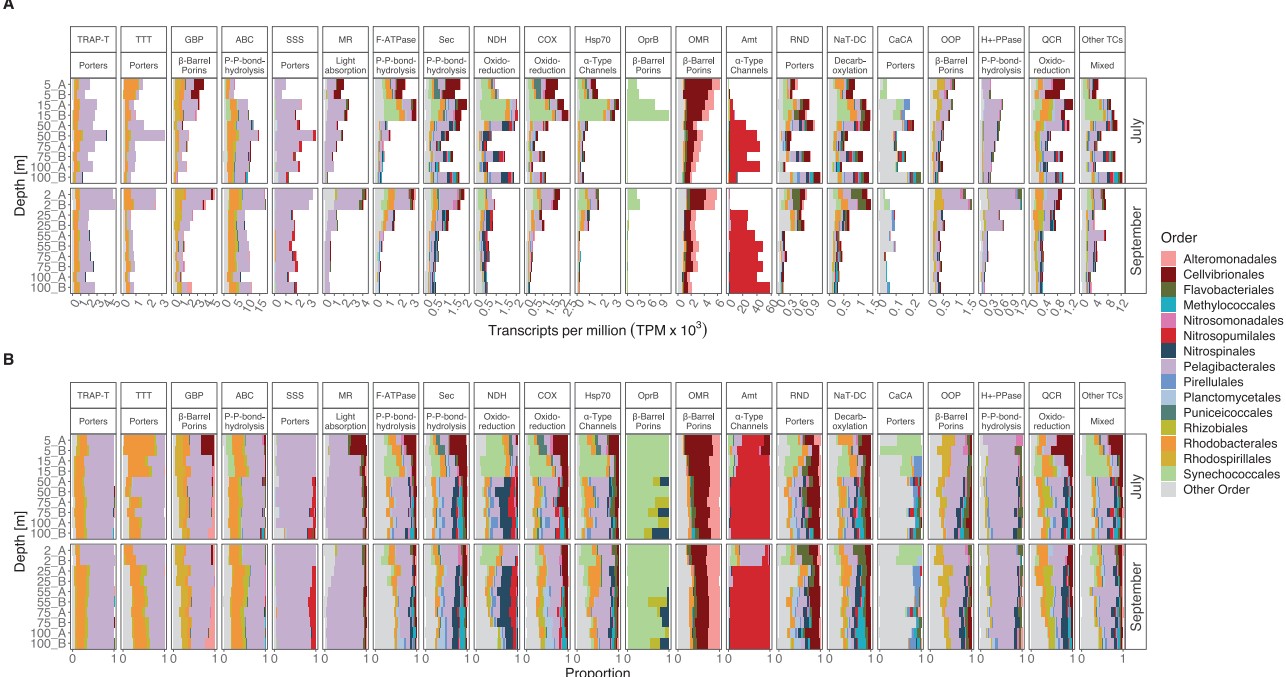

**FIG 7** Transcription of the 20 most abundant transporter families (TPs) grouped into transporter subclasses and taxonomic orders. (A) Relative transcript abundance in transcripts per million × 10³; note different scales on *y* axes. (B) Proportions of TP transcription of prokaryotic orders. Abbreviations of transporter families are the same as in the legend to Fig. 6.

family) were actively engaged in the turnover of low-molecular-weight compounds but depicted different nutritional preferences with noticeable changes between depths and sampling dates (Fig. 7; Fig. S7 and S8). Both orders showed important contributions to, e.g., tripartite ATP-independent periplasmic transporter (TRAP-T), tricarboxylate transporter (TTT), and transporters for branched-chain amino acids (PF02653) and sugars (PF00532) (Fig. 7; Fig. S7). However, pronounced differences in transcription in a variety of transporter families, including solute-sodium symporter (SSS) and proteorhodopsin (microbial rhodopsin [MR], ~3% of transcription) were noted. As anticipated, *Rhodobacterales* did not transcribe proteorhodopsins but showed a higher activity in TTT transcription in July than did *Pelagibacterales*, which contributed more in the surface in September (Fig. 7). Also, *Pelagibacterales* (e.g., "*Candidatus* genus Pelagibacter") expressed high levels of glycine betaine transporters (PF04069), whereas *Rhodobacterales* (e.g., *Rhodobacteraceae* family) transcribed more extracellular solute-binding proteins for amino acids, oligopeptides, and oligosaccharides (PF00497, PF00496, PF01547). The common osmolyte glycine betaine is readily degraded by marine bacteria, including *Pelagibacterales*, who require reduced sulfur compounds (e.g., dimethylsulfoniopropionate [DMSP] or serine) for optimal growth (41, 42). Thus, they encode high-affinity glycine betaine transporters which have a broad specificity for substrates with similar structures to glycine betaine, such as DMSP, choline, and proline (43, 44). Similarly, for *Alphaproteobacteria* in the Atlantic Ocean, the lowest relative proportions of transporter-related membrane proteins derive from *Rhodobacterales* in the mesopelagic zone (300 to 850 m), whereas *Pelagibacterales* show the highest proportions in the euphotic zone down to the mesopelagic zone (45). Taken together, these differences in transcribed transporters together with differences in the depth distribution suggest pronounced functional partitioning of nutrients between *Pelagibacterales* and *Rhodobacterales* with depth.

*Pelagibacterales* (an unknown genus along with "*Candidatus* genus Pelagibacter") largely dominated the expression of proteorhodopsin (MR transporter family), along with some expression by *Cellvibrionales* (e.g., *Porticoccaceae* and *Halieaceae* families) and *Flavobacteriales* (e.g., *Flavobacteriaceae* family) (Fig. S8). As expected, proteorhodopsin

expression was highest in the surface (especially in September), yet substantial expression by *Pelagibacterales* (by an unclassified genus within the *Pelagibacteraceae*) was also detected down to 100 m (i.e., well below the deep chlorophyll maximum at 25 m) (Fig. 7). Proteorhodopsins (PR) are light-activated proton pumps that generate a proton motive force that in turn can be utilized by ATPases to generate energy (5). These retinal-binding rhodopsin proteins are widespread in bacteria (see references 46 and 47 and references therein), including *Pelagibacterales* (SAR11 clade) (see reference 42 and references therein). The precise metabolic processes that are stimulated by PR photoheterotrophy appear to differ between prokaryotes with different life strategies (46, 47). The physiological roles of PR below the euphotic zone, if any, are still unknown. Sunlight can penetrate several hundred meters deep into the water column, where even a few remaining photons may suffice for operating PR at maintenance levels (48). Yet, it is possible that the PR expression we observed in our deep samples is constitutive, given that experimental analyses show that PR expression in some strains of taxa as divergent as vibrios (*Gammaproteobacteria*) and pelagibacters (SAR11 clade; *Alphaproteobacteria*) is independent of light (49, 50).

*Synechococcales* showed the highest transcript abundance of transporters in the subsurface Chl *a* maximum layer in July and to a lesser extent in September. Photoautotrophs which are distributed between the surface and the deep Chl *a* maximum experience sharp gradients in light and nutrients and therefore face the challenge to optimize growth by adapting their metabolic machinery (51). The transcriptional increase of ion and electron transporters (e.g., F-ATPase, Sec, NDH, COX, and Hsp70) in the Chl *a* maximum, together with the active usage of storage polysaccharides, indicates an active struggle to maintain a beneficial metabolic balance for growth, given their involvement in respiration and stress responses (52). Interestingly, *Synechococcales* (e.g., *Synechococcus*) dominated transcription of the carbohydrate-selective porin (OprB) family in July (in the Chl *a* maximum) (Fig. 7; Fig. S8). To our knowledge, such high transcription of *oprB* by this taxon across a vertical depth profile has not been reported before. In *Pseudomonas putida*, these porins are part of a high-affinity uptake system for a variety of carbohydrates, including glucose (53). Unicellular picocyanobacteria (*Prochlorococcus* and *Synechococcus*) also encode proteins with an OprB domain (54–56), and uptake of glucosyl-glycerol, sucrose, and trehalose under temperature and salinity stress has been demonstrated (see reference 57 and references therein). Thus, picocyanobacteria may acquire a variety of low-molecular-weight carbohydrates through these porins either for osmotic adjustments (57) or in response to P starvation through uptake of sugar-phosphates (55). These results provide additional support for the importance of a certain capacity for mixotrophy in picocyanobacteria (58, 59).

An important transporter family dominated by *Cellvibrionales* (e.g., *Halieaceae* and *Porticoccaceae* families) and *Alteromonadales* (e.g., *Pseudoalteromonadaceae* and *Alteromonadaceae* families) (*Gammaproteobacteria*) was the OMR family, which reached the highest relative transcription in the surface layer and decreased with depth (Fig. 7; Fig. S8). Although the relative transcription of OMR by *Cellvibrionales* (e.g., *Halieaceae* family) was more than 3-fold higher than that of *Alteromonadales* (e.g., *Alteromonadaceae*) in the July surface layer, the two taxa contributed equally in September. Transporters in the OMR family, which include TonB receptors, have broad substrate specificities and are involved in the uptake of iron-siderophore complexes, vitamin $B_{12}$, nickel complexes, colicins, and carbohydrates (60). In the North Pacific Subtropical Gyre, *Alteromonadales* increased their transcriptional activity with depth (6) and showed the highest transporter protein abundance between 300 and 800 m depth in the Atlantic Ocean (6, 25, 45). Knowledge of the ecophysiology and ecology of *Cellvibrionales* is still scarce, but they appear to reach highest relative abundances in surface waters and coastal areas (61, 62). Here, we showed that both bacterial groups dominated OMR transcription throughout the water column, suggesting a vital role of the two gammaproteobacterial orders in nutrient cycling (potentially related to carbohydrates).

Strikingly, below 15 to 25 m depth, there was a dramatic increase in ammonium transporter (Amt) family transcripts by *Thaumarchaeota* (*Nitrosopumilus* genus), accounting for up to ~79% of transporter transcripts (~22% of total transcripts) at 100 m depth (Fig. 2 and 7; Fig. S8). Other transporter families transcribed by *Thaumarchaeota* were sodium-solute (SSS) and $H^+$ or $Na^+$-translocating carboxylic acid decarboxylase (NDH). Elevated transcription of the *amt* gene by *Thaumarchaeota* has been reported from the North Pacific Subtropical Gyre from 25 to 500 m, and a high abundance of Amt transporter proteins has been reported from the Atlantic Ocean (25, 45). *Thaumarchaeota* are abundant and widespread in marine waters and sediments (29, 63, 64) and account for major proportions of ammonia oxidation in mesopelagic waters of the open ocean through the key enzyme ammonia monooxygenase (encoded by *amoABC*) (9). *Thaumarchaeota* genomes encode high-affinity ammonium transporters that enable them to outcompete ammonia-oxidizing bacteria (AOB) under low ambient ammonium concentrations (nM range) (7, 65). The variety of ABC type transporters (e.g., for amino acids, oligopeptides, phosphonates), urea transporters, and solute-sodium symporter (SSS) family transporters (45, 66, 67) enables utilization of nitrogen sources other than ammonia (68) and supports previous findings that some *Thaumarchaeota* are capable of using inorganic and organic matter to supplement their metabolism (45, 66, 69).

Collectively, these findings suggest that depth strongly shapes the transcription of transporters, similarly to CAZymes. The most pronounced shift in transcription occurred between the surface layer and 50 m depth, largely consisting of a pronounced increase in thaumarchaeal ammonium transporters. These diverse responses in different membrane transporter families, which were associated with different taxa, highlight the important contribution of a few functional groups (e.g., *Pelagibacterales*, *Rhodobacterales*, and *Thaumarchaeota*) to the turnover of nutrients (e.g., amino acids, carbohydrates, glycine betaine, DMSP, and ammonium) across depth gradients.

**Thaumarchaeal ammonia oxidation and carbon fixation.** To follow up on the exceptionally high transcriptional activity of *Thaumarchaeota* below 25 m, we analyzed marker genes for thaumarchaeal energy and carbon metabolism (Fig. S9A and B). This revealed elevated transcription of both the gene for the large subunit of archaeal ammonia monooxygenase (*amoA*) mediating the first step of nitrification and the gene for archaeal inorganic carbon fixation (*hcd*; 4-hydroxybutyryl-CoA dehydratase) in the hydroxypropionate/hydroxybutyrate (HP/HB) cycle (7). Relating this expression to the expression of the single-copy gene *radA* (70) showed that the relative investment in ammonia oxidation was high (i.e., *amoA*/*radA* ratios, 3- to 4-fold) compared to that of inorganic carbon fixation (4-hydroxybutyryl-CoA dehydratase gene/*radA* ratios, ~3-fold), especially in the upper surface layer (Fig. S9C and D).

Although archaeal ammonia oxidation is coupled with inorganic carbon fixation (71), the low energy yield from ammonia oxidation ($\Delta G = -307.35$ kJ $mol^{-1}$ $NH_3$) (72) suggests that a large quantity of ammonium needs to be oxidized per amount of $CO_2$ fixed (73). Accordingly, in soil *Thaumarchaeota*, the expression of *amoA* is severalfold higher than that of *hcd* (74), and in *Nitrosopumilus adriaticus*, nitrification resulted in a C yield per N of ~0.1 (75). Our gene expression data extend these findings on cultivated isolates to the natural marine environment and suggest that different *Thaumarchaeota* may have a distinct influence on the oceanic nitrogen compared to carbon cycles.

**Conclusion.** The overall depth distribution of prokaryotic transcription in the Gullmar Fjord agreed with findings in the Mediterranean Sea, North Pacific, and other fjords (16, 21, 25), keeping in mind that we sampled a single station in the fjord only twice. Nevertheless, our findings provide novel insights into the stratification of transcriptional activity of genes involved in transformation of DOM and nutrient uptake in natural microbial communities, which heretofore has been surveyed primarily through metagenomics (16–18). The pronounced transcriptional differences in CAZymes among distinct taxa of bacteria and archaea between samplings and over depth were notable in several respects. Given the succession often observed among phytoplankton and in relation to zooplankton grazers, we infer that ecosystem-level approaches

are urgently needed to disentangle the scale of carbon fluxes associated with bacterial utilization of phytoplankton storage polysaccharides compared to zooplankton structural polysaccharides. Moreover, building on previous studies of the genetic adaptations in nutrient scavenging with depth (see reference 45 and references therein), we suggest that expression analyses at the gene and protein levels are necessary to inform the mechanisms regulating the spatial partitioning of resources between bacterial and archaeal taxa. Characterizing and quantifying the constraints on nutrient and carbon cycling in particular depth layers, inhabited by specific sets of prokaryotic taxa, will be important for understanding how future changes in plankton dynamics influence the efficiency of the biological carbon pump.

## MATERIALS AND METHODS

**Study site and sampling.** This study was conducted in the Gullmar Fjord at station Alsbäck (58°19'22.7N, 11°32'49.0E) (Fig. 1A and B). The fjord is located on the Swedish west coast approximately 100 km north of Gothenburg. As in other fjords, the Gullmar Fjord has pronounced differences in residence time between the dynamic surface (16 to 40 days) and deep waters (46 days to a year) (76), which ultimately structures the distribution of nutrients and microbiota. We sampled a vertical depth profile spanning from surface water to 100 m depth in July and September 2017. Although not measured, we estimated that the base of the euphotic zone was between 25 and 50 m deep, as the 1% light level is often below the deep chlorophyll *a* maximum (DCM) in open ocean systems. CTD profiles were obtained prior to the selection of the desired depth layers onboard the R/V *Oscar von Sydow*. In July, we sampled water from 5, 15, 50, 75, and 100 m depths, whereas in September we took water from 2, 25, 55, 75, and 100 m depths. The different depth layers were chosen based on CTD casts and represented major transitions in physicochemical parameters (see Fig. S1A to D in the supplemental material). From all depth layers, we sampled biological duplicates (i.e., from two separate 30-liter Niskin bottles) for inorganic nutrients, chlorophyll *a* (Chl *a*), dissolved organic carbon (DOC), and metatranscriptomics (detailed information is provided in supplemental material at https://doi.org/10.6084/m9.figshare.17032961).

**Nutrients, chlorophyll *a* concentration, and prokaryotic abundance.** *In situ* nutrient concentrations were measured with a QuAAtro AutoAnalyzer and XY-3 sampler (Seal Analytics) at the Sven Lovén Centre for Marine Infrastructure in Kristineberg, Sweden. Samples of 12 ml were poured into 13-ml polystyrene tubes (Sarstedt; 55.459) prior to the analysis of $NO_3^+ + NO_2^+$, $NH_4^+$, $PO_4^{3-}$, and $SiO_2$. For total nitrogen (TotN) and total phosphorus (TotP), 20-ml aliquots were analyzed with a QuAAtro AutoAnalyzer and XY-3 sampler (Seal Analytics) with the protocol Q126 R0 Nitrate in seawater MT3B and Q125 R0 phosphate in seawater MT3A. $SiO_2$ was measured with the protocol NIOZ–Kisel Z06605_3 and $NH_4^+$ with Q033 R7 ammonia MT3B.

Samples for determining DOC concentrations were filtered through 0.2-$\mu$m syringe filters (Acrodisc syringe filters; 32 mm, 514-4131, VWR), acidified with 448 $\mu$l of 1.2 M HCl to a pH of ~2 and analyzed with a high-temperature carbon analyzer (Shimadzu TOC-5000) at the intercalibrated facility at Umeå Marine Science Centre, Umeå, Sweden. DOC concentrations were calculated as nonpurgeable organic carbon. Chl *a* concentrations were determined fluorometrically upon ethanol extraction as described previously (77).

Samples for prokaryote abundance were fixed with 1% paraformaldehyde and 0.05% glutaraldehyde (final), stained with SYBR green I nucleic acid stain (Invitrogen) (5 $\mu$M final concentration), spiked with 2 $\mu$l Flow Check high-intensity green alignment beads (Polysciences, Inc.), and analyzed with a CyFlow Cube 8 flow cytometer (Sysmex Partec) according to reference 78 (see supplemental material at https://doi.org/10.6084/m9.figshare.17032961).

**Metatranscriptomics analysis.** Water samples for metatranscriptomics were retrieved from every depth layer in biological duplicates. Approximately 3.5 liters of water was filtered through Sterivex filter units (GP 0.22 $\mu$m; EMD Millipore) onboard within ~30 min after sampling, preserved in 2 ml RNAlater (Qiagen), immediately flash frozen in liquid nitrogen until samples were transferred to a −80°C freezer at the Sven Loven Center (Kristineberg, Sweden), and later stored at −80°C at Linnaeus University (Kalmar, Sweden) until further processing. RNAlater was removed from the Sterivex filter cartridges by using a sterile 20-ml syringe and applying gentle pressure until all of the preservation solution was removed. Total RNA was extracted with an RNeasy minikit (Qiagen) according to reference 79 with some modifications as described in reference 80. In brief, total RNA was DNase treated using a TURBO DNA-free kit (ThermoFisher Scientific) and afterwards controlled for residual DNA by a 30-cycle PCR with 16S rRNA gene primers (27F and 1492R) including Milli-Q as a negative control and DNA from *Escherichia coli* as a positive control. rRNA was depleted using a RiboMinus transcriptome isolation kit and a RiboMinus concentration module (ThermoFisher Scientific). The rRNA-depleted fraction was reverse transcribed into cDNA (first and second strand) and linearly amplified (i.e., *in vitro* transcribed) using the MessageAmp II-bacterial RNA amplification kit (ThermoFisher Scientific) according to the manufacturer's instructions before library construction (TruSeq) and sequencing at the National Genome Infrastructure SciLifeLab Stockholm (Illumina HiSeq 2500 platform in rapid mode and with HiSeq SBS kit v4 chemistry to obtain 2 × 126 bp long paired-end reads) (see supplemental material at https://doi.org/10.6084/m9.figshare.17032961).

Illumina adapter sequences were removed from quality-controlled paired-end reads with Cutadapt (81) (v1.13), and reads were quality trimmed with Sickle (https://github.com/najoshi/sickle) (v1.33) in

mSystems®

paired-end mode and with Sanger quality values. Our analysis strategy was dictated by the need to identify longer mRNA fragments to improve peptide predictions, which enables the use of hmm profiles (CAZymes, peptidases, and transporters). Therefore, we decided to perform a *de novo* assembly of high-quality reads with MEGAHIT (82) (v1.1.2) and default parameters, resulting in 451,080 contigs with a minimum length of 200 bp, a maximum length of 51,124 bp, and an average contig length of 537 bp with an $N_{50}$ of 538 bp. Subsequently, open reading frames (ORFs) were determined with Prodigal (83) (v2.6.3) and default parameters. rRNAs from bacteria, archaea, and eukaryotes were predicted by running BARNAP (84) (v0.9) on the contigs with an E-value cutoff of 1e−6 and rejection of genes of <0.05 of the expected length. Subsequently, ORFs that showed an overlap with the predicted rRNAs on the contigs were removed from the downstream analysis (6,917 ORFs) (Table S1). The remaining peptides were used for blastp searches in the NCBI RefSeq protein database with DIAMOND (85) (v0.9.24) and an E-value threshold of 0.001. Thereafter, taxonomic annotation was done with MEGAN (86) (v6.12.8), using the lowest common ancestor (LCA) algorithm for asemblies with a suggested bit score of 50 and a top hit of 10%. Reads were mapped with bowtie2 (87) (v2.3.5.1) in paired-end mode to the ORFs. Subsequently, Samtools (88) (v1.9) was used to quantify transcript counts per gene. Sequencing and bioinformatics summary statistics are in Table S1.

CAZyme open reading frames (ORFs) were detected and classified with the run-dbcan program (89) (v2.0.11) and default parameters. Consensus CAZyme family classifications were filtered as follows: (i) ORFs, predicted with only one tool, were excluded if their percent identity was <50% (DIAMOND), coverage was <0.5 (HMMER), or hits were <10 (Hotpep); (ii) ORFs with positive hits from more than one prediction tool were ranked based on the number of CAZyme family classifications and their relative percent identity (DIAMOND), percent coverage (HMMER), or relative hits plus frequency (Hotpep) across all prediction tools. The highest-ranking CAZyme family classification was kept for downstream analyses. PEPs were identified using HMMER3 (90) and Pfam profiles matching hits to PEP subunit sequences in the MEROPS database (https://www.ebi.ac.uk/merops). Transporter genes were detected and classified with HMMER3 using profiles from the Transporter Classification databases (TCDB). We noticed that two PFAM domains, PF00909 and PF00654, were assigned to incorrect TCDB families, and therefore we manually reassigned the protein domain PF00909 to the "The Ammonium Channel Transporter (Amt) Family; TC 1.A.11" and protein domain PF00654 to the "The Chloride Carrier/Channel (ClC) Family; 2.A.49." This incorrect assignment is noted in the TCDB database (see supplemental material at https://doi.org/10.6084/m9.figshare.17032961).

The Archaea-specific marker genes for RadA and AmoA peptides were detected by hidden Markov models run with HMMER3 and the profiles TIGR02236 (RadA) and PF12942 (AmoA), in the TIGRFAM (v15.0) and Protein Families (Pfam) (v31.0) databases. Hits were considered valid if the score was equal to, or higher than, the recommended "gathering score" for the model. In the case of 4-hydroxybutyryl-coenzyme A (CoA) dehydratase, an alignment was constructed using MUSCLE (91) with the peptide from the characterized gene in *Nitrosopumilus maritimus* SCM1 (locustag Nmar_0207), its predicted orthologs in other *Thaumarchaeota* (7), and all peptides that contained the two conserved domains according to their PFAMs in Nmar_0207, PF11794 (4-hydroxyphenylacetate 3-hydroxylase N terminal) and PF03241 (4-hydroxyphenylacetate 3-hydroxylase C terminal) in the set of peptides derived from the genomes in the MAR database using the "gathering score" as the cutoff. A maximum likelihood phylogeny of the peptides was estimated using FastTree (92). Nmar_0207 clustered together away from predicted paralogs (peptides with a different function). To quantify Nmar_0207 ortholog expression, the peptides from the assembled metatranscriptome sequences were searched using blastp with a relaxed E value (0.0001) against the database of all MAR peptides together with Nmar_0207 predicted orthologs. Those whose closest hits were to Nmar_0207 orthologs were chosen to align them and place the new peptides on the phylogenetic tree. The packages PaPaRa (93), EPA-ng (94), and gappa (95) were used to confirm the position of the new peptides on the phylogenetic tree inside the group of 4-hydroxybutyryl-CoA dehydratase orthologs after visualization of the tree with iToL (96). The amino acid substitution model was predicted with IQ-TREE (97) (see supplemental material at https://doi.org/10.6084/m9.figshare.17032961).

**Statistics and visualization.** In total, 523,441 ORFs were predicted from the assembly, of which 6,917 were identified as rRNA and subsequently removed from further analysis. Subsetting the data set for bacteria and archaea resulted in 256,673 ORFs, which were used for downstream analyses. Principal-component analysis (PCA) was performed on Hellinger transformed raw counts according to reference 98 on ORFs with at least 5 cpm in 2 or more samples (resulting in 82,842 ORFs). Redundancy analysis (RDA) was performed on the same input data as described above. Environmental variables were standardized with the function "decostand" in vegan and selected based on pairwise Pearson correlation coefficients of <0.9 and variance inflation factors of <10. $NH_4^+$ concentrations below the detection limit of 0.2 $\mu$M were replaced with a small value (0.001) to enable the estimation of this variable in the tb-RDA. The suitability of an RDA was tested prior to analysis (gradient length of ~3.5). The model consisting of the variables temperature, DOC, Chl *a*, $NH_4^+$, $NO_3^+ + NO_2^+$, and $PO_4^+$ was significant ($P < 0.001$, $R^2_{adj} = 57\%$). Monte Carlo permutation tests showed that (i) both RDA axes were significant after Holm's correction for multiple testing ($P_{adj} < 0.006$) and (ii) the variables DOC ($P_{adj} < 0.006$), Chl *a* ($P_{adj} < 0.043$), $NH_4^+$ ($P_{adj} < 0.04$), and $NO_3^+ + NO_2^+$ ($P_{adj} < 0.043$) were significant, explaining ~12% and ~9%, ~13%, and 8% of the variation in prokaryotic community transcripts, respectively, as derived from the variance partitioning (Fig. S2). Grouping of samples was determined through hierarchical clustering based on scaled Hellinger transformed raw counts, Euclidean distances, and Ward D2 cluster criteria. The optimal number of clusters was graphically determined through Elbow and Silhouette methods. Observed richness (Chao1) and evenness (Pielou's J) were calculated with normalized counts that were scaled by ranked subsampling (SRS) according to reference 99. All statistical analyses and graphical visualizations were conducted in RStudio

(100) and predominantly with functions from the packages tidyverse (101) and vegan (102) (see supplemental material at https://doi.org/10.6084/m9.figshare.17032961).

**Data availability.** All metatranscriptome data are available at the EMBL-EBI European Nucleotide Archive repository under project accession number PRJEB42919.

## SUPPLEMENTAL MATERIAL

Supplemental material is available online only.

**FIG S1**, EPS file, 2.5 MB.
**FIG S2**, EPS file, 0.2 MB.
**FIG S3**, EPS file, 0.4 MB.
**FIG S4**, EPS file, 2.1 MB.
**FIG S5**, EPS file, 0.8 MB.
**FIG S6**, EPS file, 2.1 MB.
**FIG S7**, EPS file, 2 MB.
**FIG S8**, EPS file, 1.9 MB.
**FIG S9**, EPS file, 0.6 MB.
**TABLE S1**, PDF file, 0.2 MB.

## ACKNOWLEDGMENTS

We acknowledge the Sven Lovén Centre for Marine Sciences for their hospitality during our stay, in particular Peter Tiselius for his kind support with equipment and Hans Olsson, Lars Ljungqvist, Bengt Lundve, and Pia Engström for their assistance throughout our stays regarding laboratory equipment and nutrient analyses. We also thank Ursula Schwarz and Carl Kristenssen for operating the R/V *Oscar von Sydow* and their skillful assistance during the field sampling campaigns. A special thank goes to Sabina Arnautovic and Camilla Karlsson for their dedicated work in the laboratory.

This research was supported by a grant from the University of Gothenburg and the Royal Swedish Academy of Sciences (KVA) to B.P., C.B., and O.C.M.G. Support was also given through a grant from the Swedish Research Council VR and the marine strategic research program EcoChange to J.P. C.B. was additionally supported by HIFMB, a collaboration between the Alfred-Wegener-Institute, Helmholtz-Center for Polar and Marine Research, and the Carl-von-Ossietzky University Oldenburg, initially funded by the Ministry for Science and Culture of Lower Saxony and the Volkswagen Foundation through the Niedersächsisches Vorab grant program (grant number ZN3285). J.M.G. research was supported by the Spanish Ministry of Science and Innovation (project PID2019-110011RB-C32).

We acknowledge support from the Science for Life Laboratory, the National Genomics Infrastructure, NGI, and Uppmax (compute project SNIC 2017/7-419 and storage project SNIC 2020/16-76), Sweden, for providing assistance in massive parallel sequencing and computational infrastructure.

B.P., C.B., C.M.G.O., and J.P. designed the study. B.P., C.B., C.M.G.O., and C.P.-M. conducted the field work, retrieved samples, and processed samples in the laboratory. B.P., D.L., and J.M.G. processed metatranscriptomic data. B.P. and J.P. wrote the manuscript with contributions from all authors.

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
