## [Reviewer comments · mSystems]

Taxon-specific shifts in bacterial and archaeal transcription of dissolved organic matter cycling genes in a stratified fjord

Benjamin pontiller, Clara Pérez-Martínez, Carina Bunse, Christofer Osbeck, José González, Daniel Lundin, and Jarone Pinhassi

Corresponding Author(s): Jarone Pinhassi, Linnaeus University

Review Timeline:

Submission Date:	June 1, 2021
Editorial Decision:	June 30, 2021
Revision Received:	November 17, 2021
Accepted:	November 18, 2021

Editor: Rachel Poretsky

Reviewer(s): The reviewers have opted to remain anonymous.

Transaction Report:

DOI: <https://doi.org/10.1128/mSystems.00575-21>

June 30, 2021

Prof. Jarone Pinhassi
Linnaeus University
Centre for Ecology and Evolution in Microbial model Systems
Stuvaregatan 4
Kalmar SE-39231
Sweden

Re: mSystems00575-21 (Taxon-specific shifts in bacterial and archaeal transcription of dissolved organic matter cycling genes in a stratified fjord)

Dear Prof. Jarone Pinhassi:

Thank you for submitting your manuscript to mSystems. We have completed our review and I am pleased to inform you that, in principle, we expect to accept it for publication in mSystems. However, acceptance will not be final until you have adequately addressed the reviewer comments.

Preparing Revision Guidelines

For complete guidelines on revision requirements, please see the Instructions to Authors at <https://msystems.asm.org/sites/default/files/additional-assets/mSys-ITA.pdf>. **Submissions of a paper that does not conform to mSystems guidelines will delay acceptance of your manuscript.**

Sincerely,

Rachel Poretsky

Editor, mSystems

Journals Department
Reviewer comments:

Reviewer #1 (Comments for the Author):

This study investigated the diversity and distribution of transcripts for dissolved organic matter (DOM) cycling genes and their taxonomic affiliation in a stratified fjord located on the Swedish west coast. The authors sampled the water-column at various depths in July and September 2017 and used metatranscriptomics to characterize the prokaryotic community transcription for carbohydrate-active enzymes (CAZymes), peptidases (PEPs) and transporters (TPs).

Overall, this is an interesting and well-written manuscript that makes important contributions by revealing in situ microbial activity for genes related to organic matter metabolism at various depths of a fjord that has implications for marine ecosystems in general. The authors perform an in-depth investigation of the prokaryotic transcription of specific classes/families of CAZymes, PEPs and TPs, and find strong differences in their transcriptional profile and taxonomic affiliation between near-surface (2-15 m) and depth (50 m and below). However, there are some issues in the manuscript that the authors should address. I have described these below as well as some suggestions to improve the manuscript.

Specific comments -

In the context of the study's aim/hypothesis, the authors have performed a robust sampling effort of the Fjord water-column. However, they sampled at just one location and on two time-points. In my opinion, this puts limitations on the generalizability of the results about prokaryotic DOM metabolism in this Fjord and marine ecosystems in general. I think the authors should address this in the discussion section.

Line 118 - It is not clear here if two water samples (30L each) were collected or the total volume of the duplicates was 30L.

Line 156 - Information about cDNA synthesis from the amplified mRNA is missing - what kit/method were used?

Line 161 - I think the authors should provide more information about the sequences. For each sequenced metatranscriptome, what was the library size? the number of reads retained after QC and more importantly after removal of ribosomal RNA sequences? A supplementary table would be a good way to show this information.

Line 164 - Is this database publicly accessible?

Line 167 and Fig. 2 - How did the authors obtain a total of 82842 ORFs from the different metatranscriptomes? Details about assembling the reads with MEGAHIT and subsequent ORF prediction should be provided.

Line 169 - In my opinion, this e-value (0.001) threshold seems too liberal. I am not suggesting reanalyzing the datasets, but some justification about the choice of the e-value thresholds would be helpful.

Line 221 - I think the authors are referring to Fig. S3 here, not S5.

Lines 281 and 285 - The authors highlight trends in September in line 281 but go on to talk about July trends about the same taxa in line 285, it's a bit unclear.

Lines 346-359 - Although metabolic/transcriptional plasticity in transcribing these genes/enzymes is likely a factor in explaining the taxon-specific trends across the Fjord depth, niche-differentiation within each of the broad taxonomic groups might also be an important factor. Some microbes within an order might be more active than others at a particular depth, but this information gets masked at the higher taxonomic level (order).

Line 363 - Isn't the richness of expressed CAZymes lower in the surface from the deeper waters based on Fig. 3A?

Line 492 - This is interesting, any idea why there could be such a substantial transcription of proteorhodopsin by this bacterial group even at such depths?

Reviewer #2 (Comments for the Author):

Pontiller et al conducted a transcriptomic analysis of marine microbial communities to identify vertical patterns in taxa specific gene expression, in particular DOC metabolism gene expression. The analysis is conducted over two dates that allow for temporal comparisons in response to higher phytoplankton abundance and hydrographic properties.

The manuscript is well written, with clear goals and logical arguments. The science is sound, with a logical narrative. The paper

advances our understanding of how DOC related gene expression, particularly in relation to enzymes for degrading HMW-DOM, are differentially throughout the water column. The ancillary hydrographic and nutrient data provide good context for the data. The figures are particularly well done, especially in the way that they both capture the complexity of transcription across gene function and taxonomy classes in combination with depth, while also clearly communicating the important trends. The authors also do a good job not only convincingly showing the different gene expression trends, but they clearly explain the functions of these genes and how they might fit into the ecology of the stratified fjord system.

I have only a couple major comments/suggestions and some detailed edits:

Why were the reads assembled? While this is common for eukaryotic metatranscriptomes, it is less so for prokaryotic samples. The main difference being that the higher diversity of prokaryotic taxa and genes increases the chance that assemblies of short reads for transcriptomics can result in increased prevalence of chimeras. Please explain the rationale for doing the assembly and any steps that were taken to control for chimeric sequences. Also, please describe how many of the reads didn't then map to a contig. What was the fate of these reads?

The relative nature of the dataset is often adequately addressed. However, in certain parts of the manuscript this is overlooked. For example, on L330, the authors state that the dominance of thaumarchaeota amt transport highlights a transition from organic matter utilization to inorganic nutrient uptake. Alternatively, absolute abundance of organic processing gene expression could be similar at depth, but high expression of Thaum amts makes it look like the bacterial organic genes are less expressed. A discussion of how this absolute versus relative abundance issue might affect your interpretations is thus warranted. This could be done by just acknowledging it before the next paragraph (L346), in which you show that the taxa specific transcription changes with depth, which I think does a great deal to support the proposed model of changes in organic processing with depth.

Detailed Comments:

L363 and 365. These statements are confusing. From figure 3 it looks like richness is low at the surface and increases with depth. Indeed this seems to be what L365 confirms as well.

L148. Please describe how you removed the RNAlater from your sterivex filters.

L394 Incomplete sentence.

L473. A bit misleading wording given that the vast majority of roseobacters do not contain PR genes.

L492. Do they have any info on where the base of the euphotic zone is? Maybe this could be indicated on one of the plots in Figure 1.

mSystems00575-21R1 (Taxon-specific shifts in bacterial and archaeal transcription of dissolved organic matter cycling genes in a stratified fjord)

Please find below our responses to the reviewer comments in boldface:

Reviewer comments:

Reviewer #1 (Comments for the Author):

This study investigated the diversity and distribution of transcripts for dissolved organic matter (DOM) cycling genes and their taxonomic affiliation in a stratified fjord located on the Swedish west coast. The authors sampled the water-column at various depths in July and September 2017 and used metatranscriptomics to characterize the prokaryotic community transcription for carbohydrate-active enzymes (CAZymes), peptidases (PEPs) and transporters (TPs).

Overall, this is an interesting and well-written manuscript that makes important contributions by revealing in situ microbial activity for genes related to organic matter metabolism at various depths of a fjord that has implications for marine ecosystems in general. The authors perform an in-depth investigation of the prokaryotic transcription of specific classes/families of CAZymes, PEPs and TPs, and find strong differences in their transcriptional profile and taxonomic affiliation between near-surface (2-15 m) and depth (50 m and below). However, there are some issues in the manuscript that the authors should address. I have described these below as well as some suggestions to improve the manuscript.

We would like to thank the reviewer for taking the time to review our manuscript and for the constructive and positive feedback. We are grateful for the suggestions to improve the quality of the manuscript and have addressed the reviewer's concerns in the text below in bold font. Also, we indicated line numbers where necessary.

Specific comments -

In the context of the study's aim/hypothesis, the authors have performed a robust sampling effort of the Fjord water-column. However, they sampled at just one location and on two time-points. In my opinion, this puts limitations on the generalizability of the results about prokaryotic DOM metabolism in this Fjord and marine ecosystems in general. I think the authors should address this in the discussion section.

We thank the reviewer for the constructive comment. While we have tried to put our work in a broader context, we appreciate the suggestion to acknowledge the limitation of a single sample location and a few sampling time points. We do agree with the reviewer that especially the limited time resolution prevents us from extrapolating our results to other seasons. Nevertheless, our data consisting of two different time points show that a comparison with other open ocean systems is feasible but extrapolating the findings to other seasons especially in the Gullmar Fjord is limited.

In order to address the reviewer's concern, we have now added text at two locations telling the limitations of this study more clearly.

First, at the beginning of the Results and Discussion section, we write (L282) "The Chl a concentrations measured here emphasize the mesotrophic nature of the Gullmar Fjord (22). The ~2-fold difference between Chl a in July and September was likely due to elevated grazing pressure in July (personal observation), in line with previous observations (22). In terms of physicochemical water column characteristics and nutrient dynamics during the time of sampling, the Gullmar Fjord compares to other fjord systems (19, 20). Still, we recognize that our current study is limited to a single location in the Gullmar Fjord, which, although sampled at two ecologically relevant time points, does not cover the full range of (a)biotic gradients along the extension of the fjord or how they change during a full year in this or other fjord systems."

Next, in the Conclusion (L635) we complemented the first sentence as follows: "The overall depth distribution of prokaryotic transcription in the Gullmar Fjord agreed with findings in the Mediterranean Sea, North Pacific, and other fjords (16, 21, 52), keeping in mind that we sampled a single station in the fjord only twice. Nevertheless, our findings provide novel insights into the stratification of transcriptional activity of genes..."

Line 118 - It is not clear here if two water samples (30L each) were collected or the total volume of the duplicates was 30L.

We have re-written this sentence for clarity:

L120 "From all depth layers, we sampled biological duplicates (i.e., from two separate 30L Niskin bottles) for inorganic nutrients, chlorophyll a (Chl a), dissolved organic carbon (DOC), and metatranscriptomics. Detailed information is provided in Supplementary File 1."

Line 156 - Information about cDNA synthesis from the amplified mRNA is missing - what kit/method were used?

This information has now been included in the Material and Methods section (L161): "The rRNA-depleted fraction was reverse transcribed into cDNA (first and second strand) and linearly amplified (i.e., *in vitro* transcribed) using the MessageAmp II-Bacteria RNA Amplification Kit (ThermoFisher Scientific) according to manufacturer's instructions before library construction (TruSeq) and sequencing at the National Genome Infrastructure, SciLifeLab Stockholm (Illumina HiSeq 2500 platform in rapid mode and with HiSeq SBS kit v4 chemistry to obtain 2×126 bp long paired-end reads) (see Supplemental Material at [<https://doi.org/10.6084/m9.figshare.17032961>])."

Line 161 - I think the authors should provide more information about the sequences. For each sequenced metatranscriptome, what was the library size? the number of reads retained

after QC and more importantly after removal of ribosomal RNA sequences? A supplementary table would be a good way to show this information.

We are grateful for the suggestions to include the missing information. We have now included a Supplementary Table (Table S1) showing sequencing and bioinformatics summary statistics. We refer to Table S1 in lines L181 and L188.

During the process of gathering this information, we noticed that the rRNA filtration step had not been correctly described in the Material and Methods text. Previously, we had written in the M&M section that the rRNA gene transcript filtration was done with ERNE against an in-house database. However, early on in the bioinformatic processing, we realized that this step did not remove any rRNA reads before the assembly with MEGAHIT (due to a missing coding step in our workflow). Thus, the assembly was done on quality-filtered reads, including rRNA reads, and rRNAs from bacteria, archaea, and eukaryotes were identified with BARNAP (which is essentially a collection of HMM profiles for stable rRNAs) and excluded from downstream analysis afterward. We have now corrected the description of how rRNAs were removed in (L177). (Please see also the response to the comment for Line 167 below).

Line 164 - Is this database publicly accessible?

As described in response to the comment L161 above, we actually did not use this database. We now describe in the Material and Methods how ribosomal RNA transcripts were detected with BARNAP and removed from downstream analyses (L177). We thus no longer refer to this as an in-house database in the manuscript (but if the reviewer is interested it is accessible upon request).

Line 167 and Fig. 2 - How did the authors obtain a total of 82842 ORFs from the different metatranscriptomes? Details about assembling the reads with MEGAHIT and subsequent ORF prediction should be provided.

We have added additional information in the form of a Supplementary Table and text in the Material & Methods section. These sections read as follows:

L170 “Our analysis strategy was dictated by the need to identify longer mRNA fragments to improve peptide predictions, which enables the use of hmm profiles (CAZymes, peptidases, and transporters). Therefore, we decided to perform a de-novo assembly of high-quality reads with MEGAHIT (29) (v1.1.2) and default parameters, resulting in 451080 contigs with a minimum length of 200 bp, a maximum length of 51124 bp, and an average contig length of 537 bp with an N50 of 538 bp. Subsequently, open reading frames (ORFs) were determined with Prodigal (30) (v2.6.3) and default parameters. Ribosomal RNAs from bacteria, archaea, and eukaryotes were predicted by running BARNAP (31) (v0.9) on the contigs with an e-value cutoff of 1e-6 and rejection of genes <0.05 of the expected length. Subsequently, ORFs that showed an overlap with the predicted rRNAs on the contigs were removed from the downstream analysis (6917 ORFs) (Table S1).”.

L231 “In total, 523441 ORFs were predicted from the assembly, of which 6917 were identified as rRNAs and subsequently removed from further analysis. Subsetting the dataset for bacteria and archaea resulted in 256673 ORFs, which were used for downstream analyses. Principal component analysis (PCA) was performed on Hellinger transformed raw counts according to (45) on ORFs with at least 5 counts per million (cpm) in 2 or more samples (resulting in 82842 ORFs).”

Line 169 - In my opinion, this e-value (0.001) threshold seems too liberal. I am not suggesting reanalyzing the datasets, but some justification about the choice of the e-value thresholds would be helpful.

We do agree with the reviewer that this e-value seems somewhat liberal for a functional cutoff. However, we only used DIAMOND to perform a taxonomic annotation of ORFs. Hence, the actual cutoff was dictated by the post-processing step with MEGAN, in which we used the suggested bit-score of 50, a top hit of 10%, and the lowest common ancestor (LCA) algorithm. Therefore, the chosen DIAMOND e-value does not influence the outcome of the taxonomic annotation.

We re-wrote this section for clarity. The section reads now as follows:

L181 “The remaining peptides were used for blastp searches in the NCBI Refseq protein database with DIAMOND (32) (v0.9.24) and an e-value threshold of 0.001; thereafter taxonomic annotation was done with MEGAN (33) (v6.12.8). The longReads lcaAlgorithm for assemblies, a suggested bit-score of 50, a top hit of 10%, and the lowest common ancestor (LCA) algorithm were chosen.”

Line 221 - I think the authors are referring to Fig. S3 here, not S5.

We thank the reviewer for pointing this out. We have now corrected this mistake and refer to Figure S2 (Previously Figure S3) (L247).

Lines 281 and 285 - The authors highlight trends in September in line 281 but go on to talk about July trends about the same taxa in line 285, it's a bit unclear.

We have now re-written this paragraph for improved clarity and flow. The paragraph now reads as follows:

L307 “Alpha- and Gammaproteobacteria accounted for high portions (around 25-30%) of the prokaryotic community transcription in the surface waters and maintained elevated transcription (~11%) throughout the water column (Fig. 2C). Cyanobacteria (primarily Synechococcus) were most active in the upper water column (reaching ~15% of community transcription). These patterns are largely in agreement with marine metagenomic and metatranscriptomic surveys from the open ocean, which typically report a dominance of Picocyanobacteria (Prochlorococcus), Alphaproteobacteria (primarily the SAR11 clade), and Bacteroidetes in the upper water column (6, 16, 52). The relatively high activity of Verrucomicrobia in July (reaching ~45% of community transcription) is in line with a report from the Baltic

Sea, where 16S rRNA gene analyses show that this taxon reaches higher relative abundances during summer coinciding with a dominance of *Cyanobacteria* (53). Metagenomic analysis of Verrucomicrobia shows that this taxon carries an extensive repertoire of CAZymes (e.g., alpha- and beta-galactosidases, xylanases, fucosidases, agarases, and endoglucanases), suggesting that they play a vital role in the degradation of phytoplankton-derived DOM and complex polysaccharides such as fucoidans (16, 54, 55).”.

Lines 346-359 - Although metabolic/transcriptional plasticity in transcribing these genes/enzymes is likely a factor in explaining the taxon-specific trends across the Fjord depth, niche-differentiation within each of the broad taxonomic groups might also be an important factor. Some microbes within an order might be more active than others at a particular depth, but this information gets masked at the higher taxonomic level (order).

We realize this could be important in many ways. We have therefore now included two additional Supplementary Figures showing the expression of CAZymes and transporters covering the taxonomic levels from class to genus levels resolution. We thereby provide additional information on the relative contribution of detected genera and families within an order to address the potential of niche differentiation within the order level (although it should be kept in mind that the taxonomic resolution differs between orders/families/genera depending on their representation in sequence databases). The main findings from this new information have been incorporated in the Results and Discussion section.

E.g., L428: “Already at the order level, there were pronounced differences in the transcription of the most abundant CAZyme families between months and depth (Fig. 5 and S5). For instance, Cellvibrionales (e.g., *Haliaceae* family) and Rhodobacterales (e.g., *Rhodobacteraceae* family) dominated transcription of glycoside hydrolase family 19 (GH19) in the surface layer, whereas Methylococcales (e.g., *Methylococcaceae* family) dominated below 50 m depth (Fig. S5 and S6).”,

and at L448: “Synechococcales (e.g., the genus *Synechococcus*) dominated transcription of GH13 at 15 m in July, with some contribution also from Alphaproteobacteria (e.g., Rhodobacterales and *Pelagibacteraceae* family) (Fig. 5 and S6).”,

and L508: “Our findings on membrane transporter transcription indicated that Pelagibacterales (e.g., *Candidatus* genus *Pelagibacter*) and Rhodobacterales (e.g., *Rhodobacteraceae* family) were actively engaged in the turnover of low molecular weight compounds but depicted different nutritional preferences with noticeable changes between depths and sampling dates (Fig. 7, S7, and S8).”.

Line 363 - Isn't the richness of expressed CAZymes lower in the surface from the deeper waters based on Fig. 3A?

We thank the reviewer for pointing this out. We have now corrected the sentence accordingly (L396)

Line 492 - This is interesting, any idea why there could be such a substantial transcription of proteorhodopsin by this bacterial group even at such depths?

Given that experimental results studying the effect of PR on bacterial growth under light and dark conditions gives different results for different taxa (e.g., no significant differences in growth or expression in pelagibacters and vibrios but stimulation in some other Gammaproteobacteria and Bacteroidetes), it is difficult to speculate on the role of PR in the deep Ocean. However, we have added two sentences to the existing section in an attempt to address the reviewer's question to the best of our knowledge. The two sentences read as follows:

L541 "Proteorhodopsins (PR) are light-activated proton pumps that generate a proton motive force that in turn can be utilized by ATPases to generate energy (5). These retinal-binding rhodopsin proteins are widespread in bacteria (73, 74 and references therein), including Pelagibacterales (SAR11 clade) (69 and references therein). The precise metabolic processes that are stimulated by PR photoheterotrophy appear to differ between prokaryotes with different life strategies (73, 74). The physiological roles of PR below the euphotic zone, if any, are still unknown. Sunlight can penetrate several hundred meters deep into the water column where even a few remaining photons may suffice for operating PR at maintenance levels (76). Yet, it is possible that the PR expression we observed in our deep samples is constitutive, given experimental analyses show that PR expression in some strains of taxa as divergent as vibrios (Gammaproteobacteria) and pelagibacters (SAR11 clade; Alphaproteobacteria) is independent of light (76, 77)."

Reviewer #2 (Comments for the Author):

Pontiller et al conducted a transcriptomic analysis of marine microbial communities to identify vertical patterns in taxa specific gene expression, in particular DOC metabolism gene expression. The analysis is conducted over two dates that allow for temporal comparisons in response to higher phytoplankton abundance and hydrographic properties.

The manuscript is well written, with clear goals and logical arguments. The science is sound, with a logical narrative. The paper advances our understanding of how DOC related gene expression, particularly in relation to enzymes for degrading HMW-DOM, are differentially throughout the water column. The ancillary hydrographic and nutrient data provide good context for the data. The figures are particularly well done, especially in the way that they both capture the complexity of transcription across gene function and taxonomy classes in combination with depth, while also clearly communicating the important trends. The authors also do a good job not only convincingly showing the different gene expression trends, but they clearly explain the functions of these genes and how they might fit into the ecology of the stratified fjord system.

We are grateful to the reviewer's very positive feedback and for taking the time to review our manuscript.

I have only a couple major comments/suggestions and some detailed edits:

Why were the reads assembled? While this is common for eukaryotic metatranscriptomes, it is less so for prokaryotic samples. The main difference being that the higher diversity of prokaryotic taxa and genes increases the chance that assemblies of short reads for transcriptomics can result in increased prevalence of chimeras.

Please explain the rationale for doing the assembly and any steps that were taken to control for chimeric sequences.

Also, please describe how many of the reads didn't then map to a contig.

What was the fate of these reads?

Our analysis strategy allowed us to increase the length of the reads by assembling them into longer contigs, which improve open reading frame predictions and subsequent functional annotation of fragments via hmm profile searches. To our knowledge, this is a commonly used and generally accepted approach in microbial ecology. Also, we believe that it is virtually impossible to quantify chimeras in de novo assemblies of metatranscriptomes without having genomes or MAGs from the same sample available. Chimeras are likely to impair samples with highly similar compositions such as human cell tissues and/or cross-kingdom comparisons. We do think that the increased fragment length of assembled reads outweighs the potential limitations of chimeras, given that we focused exclusively on bacteria and archaea.

We have now explained our rationale behind the chosen analysis in the Material and Methods section:

L170 "Our analysis strategy was dictated by the need to identify longer mRNA fragments to improve peptide predictions, which enables the use of hmm profiles (CAZymes, peptidases, and transporters). Therefore, we decided to perform a de-novo assembly of high-quality reads with MEGAHIT (29) (v1.1.2) and default parameters, resulting in 451080 contigs with a minimum length of 200 bp, a maximum length of 51124 bp, and an average contig length of 537 bp with an N50 of 538 bp."

We have now included a Supplementary Table (Table S1) showing sequencing and bioinformatics summary statistics. We refer to Table S1 in lines L181 and L188.

The relative nature of the dataset is often adequately addressed. However, in certain parts of the manuscript this is overlooked. For example, on L330, the authors state that the dominance of thaumarchaeota amt transport highlights a transition from organic matter utilization to inorganic nutrient uptake. Alternatively, absolute abundance of organic processing gene expression could be similar at depth, but high expression of Thaum amts makes it look like the bacterial organic genes are less expressed. A discussion of how this absolute versus relative abundance issue might affect your interpretations is thus warranted. This could be done by just acknowledging it before the next paragraph (L346), in which you show that the taxa specific transcription changes with depth, which I think does a great deal to support the proposed model of changes in organic processing with depth.

We thank the reviewer for pointing this out. We now clearly state that the analysis is based on relative abundances and acknowledge the limitations of relative omics data at the beginning of the section starting at:

L358: “These pronounced patterns in depth distributions, in turn, inspired further analysis of the allocation of transcriptional efforts to CAZymes, PEPs, and TPs among the dominant prokaryotic orders (Fig. 3), in an attempt to mitigate the limitation of relative metatranscriptomic data (i.e., the abundance of gene transcripts calculated as percent of the sequence library) to disentangle the extent or direction of change in gene transcription of complex natural communities with depth. This highlighted the disproportionate contribution of Thaumarchaeota in nutrient uptake over polymer degradation at depth.”.

Detailed Comments:

L363 and 365. These statements are confusing. From figure 3 it looks like richness is low at the surface and increases with depth. Indeed this seems to be what L365 confirms as well.

We thank the reviewer for pointing out this mistake. We have now corrected the sentence starting L396.

L148. Please describe how you removed the RNAlater from your sterivex filters.

This information was provided in the Supplementary Material but not in the Material and Methods. We have now added a sentence to the Material and Methods section as requested by the reviewer. The sentence reads as follows:

L153 “RNAlater was removed from the sterivex filter cartridges by using a sterile 20 mL syringe and applying gentle pressure until all of the preservation solution was removed.”.

L394 Incomplete sentence.

Thank you for pointing this out. We have now completed the sentence which reads as follows:

L428 “Already at the order level, there were pronounced differences in the most abundant CAZyme families between months and with depth (Fig. 5 and S5).”.

L473. A bit misleading wording given that the vast majority of roseobacters do not contain PR genes.

We re-wrote the sentence to avoid misleading wording as suggested by the reviewer. The sentence reads as follows:

L517 “As anticipated, Rhodobacterales did not transcribe proteorhodopsins but showed a higher activity in TTT transcription in July compared to Pelagibacterales, which contributed more in the surface in September (Fig. 7).”.

L492. Do they have any info on where the base of the euphotic zone is? Maybe this could be indicated on one of the plots in Figure 1.

We thank the reviewer for this suggestion. Unfortunately, the used CTD device was not equipped with a Photosynthetically Active Radiation (PAR) Sensor. Therefore, we can't determine the base of the euphotic zone. However, typically the 1% light level is often below the DCM layer in open ocean systems. Hence, one could assume that the base of the euphotic zone was roughly between 25 and 50 meters.

We have added a sentence in M&M telling the estimated depth of the euphotic zone.

L112: “We sampled a vertical depth profile spanning from surface water to 100 m depth in July and September 2017. Although not measured, we estimated that the base of the euphotic zone was between 25 and 50 m deep, as the 1% light level is often below the deep chlorophyll a maximum (DCM) in open ocean systems.”

November 18, 2021

Prof. Jarone Pinhassi
Linnaeus University
Centre for Ecology and Evolution in Microbial model Systems
Stuvaregatan 4
Kalmar SE-39231
Sweden

Re: mSystems00575-21R1 (Taxon-specific shifts in bacterial and archaeal transcription of dissolved organic matter cycling genes in a stratified fjord)

Dear Prof. Jarone Pinhassi:

Thank you for comprehensively addressing the reviewer comments. Your manuscript has been accepted, and I am forwarding it to the ASM Journals Department for publication. For your reference, ASM Journals' address is given below. Before it can be scheduled for publication, your manuscript will be checked by the mSystems senior production editor, Ellie Ghatineh, to make sure that all elements meet the technical requirements for publication. She will contact you if anything needs to be revised before copyediting and production can begin. Otherwise, you will be notified when your proofs are ready to be viewed.

Publication Fees:

We recognize that the video files can become quite large, and so to avoid quality loss ASM suggests sending the video file via <https://www.wetransfer.com/>. When you have a final version of the video and the still ready to share, please send it to Ellie Ghatineh at eghatineh@asmusa.org.

Sincerely,

Rachel Poretsky
Editor, mSystems

Journals Department
Fig. S5: Accept
Fig. S6: Accept
Fig. S4: Accept
Fig. S2: Accept
Fig. S7: Accept
Fig. S3: Accept
Table S1: Accept
Fig. S9: Accept
Fig. S8: Accept
Fig. S1: Accept